# Integrative Analysis of DNA Methylation and microRNA Reveals GNPDA1 and SLC25A16 Related to Biopsychosocial Factors Among Taiwanese Women with a Family History of Breast Cancer

**DOI:** 10.3390/jpm15040134

**Published:** 2025-03-30

**Authors:** Sabiah Khairi, Chih-Yang Wang, Gangga Anuraga, Fidelia Berenice Prayugo, Muhamad Ansar, Mohammad Hendra Setia Lesmana, Lalu Muhammad Irham, Chen-Yang Shen, Min-Huey Chung

**Affiliations:** 1School of Nursing, College of Nursing, Taipei Medical University, Taipei City 11031, Taiwan; d432110004@tmu.edu.tw; 2Ph.D. Program for Cancer Molecular Biology and Drug Discovery, College of Medical Science and Technology, Taipei Medical University, Taipei City 11031, Taiwan; chihyang@tmu.edu.tw; 3Graduate Institute of Cancer Biology and Drug Discovery, College of Medical Science and Technology, Taipei Medical University, Taipei City 11031, Taiwan; 4Department of Statistics, Faculty of Science and Technology, Universitas PGRI Adi Buana, Surabaya 60234, Indonesia; g.anuraga@unipasby.ac.id; 5Chang Gung Medical Education Research Centre (CG-MERC), Chang Gung Memorial Hospital, Taoyuan City 33302, Taiwan; fideliaprayugo@gmail.com; 6School of Medicine, Chang Gung University, Taoyuan City 33302, Taiwan; 7Ph.D. Program in the Clinical Drug Development of Herbal Medicine, Taipei Medical University, Taipei City 110301, Taiwan; muhamadanshar919@gmail.com; 8Department of Mental Health and Community, Faculty of Medicine, Public Health, and Nursing, Universitas Gadjah Mada, Yogyakarta 55281, Indonesia; hendralesmana090294@gmail.com; 9Faculty of Pharmacy, Universitas Ahmad Dahlan, Yogyakarta 55164, Indonesia; lalu.irham@pharm.uad.ac.id; 10Institute of Biomedical Sciences, Academia Sinica, Taipei City 11529, Taiwan; 11Master Program in Clinical Genomics and Proteomics, School of Pharmacy, Taipei Medical University, Taipei City 11031, Taiwan; 12College of Public Health, China Medical University, Taichung City 406040, Taiwan; 13Department of Nursing, Shuang Ho Hospital, Taipei Medical University, New Taipei City 23561, Taiwan

**Keywords:** integrative analysis, DNA methylation, microRNA, biopsychosocial factors, family history of breast cancer

## Abstract

Biopsychosocial factors, including family history, influence the development of breast cancer. Malignancies in women with a family history of breast cancer may be detectable based on DNA methylation and microRNA. **Objectives**: The present study extended an integrative analysis of DNA methylation and microRNA to identify genes associated with biopsychosocial factors. **Methods**: We identified 3060 healthy women from the Taiwan Biobank and included 32 blood plasma samples for analysis of biopsychosocial factors and epigenetic changes. GEO databases and bioinformatics approaches were used for the identification and validation of potential genes. **Results**: Our integrative analysis revealed GNPDA1 and SLC25A16 as potential genes. Age, a family history of cancer, and alcohol consumption were associated with GNPDA1 and SLC25A16 based on the current data set and the GEO data set. GNPDA1 and SLC25A16 exhibited significant expression in breast cancer tissues based on UALCAN analysis, where they were overexpressed and underexpressed, respectively. Through a MethSurv analysis, GNPDA1 hypomethylation and SLC25A16 hypermethylation were associated with poor prognoses in terms of overall survival in breast cancer. Moreover, through a MetaCore functional enrichment analysis, GNPDA1 and SLC25A16 were associated with the BRCA1, BRCA2, and pro-oncogenic actions of the androgen receptor in breast cancer. Further, GNPDA1 and SLC25A16 were enriched in known targets of approved cancer drugs as potential genes associated with breast cancer. **Conclusions**: These two genes might serve as biomarkers for the early detection of breast cancer, especially for women with a family history of breast cancer.

## 1. Introduction

Breast cancer has a complex etiology and complex mechanisms. To reduce breast cancer-related morbidity and mortality, individuals who are at risk of breast cancer should be identified, and early intervention strategies should be employed [1,2,3]. Biopsychosocial factors play a role in the etiology of all diseases. Biopsychosocial factors are associated with the development of breast cancer, including sociodemographic [4,5] and reproductive characteristics [6], lifestyle, psychological stress, coping ability, and a family history of breast cancer (FHBC) [4,5]. Family history is a well-established biological risk factor for breast cancer that contributes significantly to the risk of the disease (odds ratio: 1.71, 95% CI: 1.59–1.84) [7]. A study involving a large patient cohort found that women who have two or more relatives with a history of breast cancer are 2.5 times more likely to develop breast cancer (95% CI: 1.83–3.47) [8]. The data suggest that FHBC is one of the most important screening criteria for identifying individuals with breast cancer risk. In addition to inherited gene mutations, a number of demographic and other biopsychosocial factors have been demonstrated to be related to increased gene alterations in breast cancer, including age [9], alcohol consumption [10], smoking [11], having borne offspring (nulliparous) [6], and a family history of breast cancer [8]. While inherited gene mutations play a key role in familial breast cancer risk, epigenetic modifications, such as DNA methylation and microRNA (miRNA) regulation, also contribute to cancer susceptibility. These epigenetic mechanisms can be influenced by both genetic predisposition and biopsychosocial factors, potentially altering gene expression and increasing breast cancer risk in individuals with FHBC.

DNA methylation and miRNA are two key epigenetic processes that regulate cellular transformation and cancer [12,13,14]. Breast cancer has been linked to DNA methylation alterations, which are epigenetic modifications that are persistent yet modifiable. Disease risk factors may disrupt epigenetic regulation in order to modulate their effect on disease propensity [1,11]. Moreover, the expression of miRNA is controlled by the DNA methylation of their promoters, and abnormal miRNA promoter methylation patterns are associated with cancer [14]. Considering the risk factors related to FHBC, Wu et al. [8] conducted a study to determine specific differentially methylated regions that are related to breast cancer susceptibility in girls with and without FHBC. In that study, numerous CpG sites frequently appeared in the ESR1 amplicon. The mean methylation levels were higher in girls with FHBC than in those without FHBC [8]. While previous studies have identified differentially methylated regions associated with breast cancer risk, the interplay between DNA methylation, miRNA expression, and biopsychosocial factors in women with FHBC remains largely unexplored. Understanding this relationship could provide deeper insights into potential early biomarkers for breast cancer risk.

Previous studies have found associations between DNA methylation-based risk variables and 18 traditional breast cancer risk factors in disease-free women [9], smoking and breast cancer DNA methylation patterns [11], and allele-specific DNA methylation among breast cancer family history [8]. These three studies have only discovered the relationship between DNA methylation and breast cancer risk factors. However, in epigenetics, DNA methylation and miRNA play a critical role, in which a dysfunctional miRNA promoter methylation pattern is associated with cancer-related miRNA expression. Therefore, this study aims to identify potential biomarkers by exploring the relationship between DNA methylation, miRNA expression, and biopsychosocial factors in Taiwanese women with FHBC. While our research focuses on healthy individuals, the integration of biopsychosocial factors and external datasets provides meaningful insights into potential early molecular alterations associated with breast cancer risk. By integrating epigenetic and bioinformatics analyses, we seek to identify potential biomarkers that may contribute to breast cancer susceptibility.

## 2. Materials and Methods

### 2.1. Study Design

In this study, we conducted several analyses to reveal potential genes, as shown in Figure 1. First, the analysis of biopsychosocial factors between FHBC and non-FHBC was investigated using the chi-square and *t*-test. Meanwhile, DNA methylation and miRNA expression profiling were conducted utilizing the Illumina Infinium HD Methylation microarray and TruSeq Small RNA Library Preparation Kits. Analysis of differentially methylated (DEM) and differential expressed microRNA (DEmiRNA) was performed using the Linear Model for Microarray (LIMMA) package version 3.62.2 and Differential Gene Expression Analysis of RNA-seq 2 (DESeq2) algorithms from the Bioconductor software package 3.18 and R version 4.3.0. Second, two Gene Expression Omnibus (GEO) data sets were used to identify potential genes related to biopsychosocial factors. Third, to fully understand the molecular pathways associated with breast cancer progression and to determine functional enrichment strategies for identifying new prognostic markers of this complex disease, a thorough investigation is required into the potential genes associated with breast cancer.

Using the TCGA databases through the UALCAN analysis platform, we compared the messenger RNA (mRNA) expression levels of potential genes in breast cancer and normal tissues to assess their potential as biomarkers. Furthermore, to investigate the relationship between potential genes and prognosis, a survival analysis was performed using the MethSurv web tool to obtain the distribution of genetic alterations among breast cancer patients. Univariate and multivariate analyses were performed using Cox proportional hazards models. Moreover, we used cBioPortal, GeneMANIA, Enrichr database, and MetaCore to determine the genetic alterations, gene interactions, gene ontology, and signaling pathways connected to GNPDA1 and SLC25A16 in patients with breast cancer. Herein, we also identified proteins in direct PPI with these encoded by two potential genes to guide the selection of drug target genes to drive drug repurposing for breast cancer using the STRING and drug databases.

### 2.2. Study Population

The study population includes participants of the Taiwan Biobank from 2008 to 2018. Population-based screening research was conducted to investigate breast cancer risk factors among healthy women. The study participants were mainly from the Taiwanese population. Women aged 35–75 years with and without FHBC were considered eligible for enrollment. A total of 3060 healthy women were included in this study. Participants were divided into FHBC (1676) and non-FHBC (1384) groups. Regarding the methylation and miRNA expression profiling, the selection for individuals in the case group was based on stringent criteria, focusing on women with two or more first-degree relatives with breast cancer (n = 72 of 1676). However, due to the high cost and resource-intensive nature of DNA methylation analysis, we selected a representative subset of 16 of 72 individuals who have two or more first-degree relatives with breast cancer. This selection criterion was designed to focus on identifying potential early molecular signatures associated with family history as a risk factor.

Regarding the miRNA expression profiling, the participants included in this experiment were selected from the same group as those in the methylation experiment. However, due to sample availability and quality control constraints, miRNA expression could not be profiled for all 32 individuals. Hence, 16 participants were selected for miRNA expression analysis (n = 8 for each group). Informed consent was obtained from all subjects involved in the study.

### 2.3. Analysis of Biopsychosocial Factors

In the Taiwan Biobank, a self-reporting technique is mainly used to acquire phenotypic data. Data on demographic characteristics, lifestyle, exposure to the environment, family history, and health condition were collected using a structured questionnaire in an interview [15]. The data were collected using the questionnaire which included three main factors. Initially, biological factors involving age, body mass index (BMI), reproductive characteristics (age at menarche, pregnancy history, age at first pregnancy, childbirth experience, age at first live birth, and breastfeeding practices), as well as first-degree relatives with breast cancer (mother and/or sister) are included. Secondly, psychological and behavioral characteristics include alcohol use (never, occasional, not recently, recently), smoking behaviors (yes/no), exposure to secondhand smoke (yes/no), and participation in exercise (yes/no). Finally, social factors encompass marital status (unmarried/married), level of education (primary school/high school and above), and dependent status (yes/no). The distribution of biopsychosocial factors between the FHBC and non-FHBC groups was compared using the Chi-square and Fisher tests for categorical variables and an independent *t*-test for continuous data.

### 2.4. DNA Methylation and miRNA Quantification

The Illumina Infinium HD Methylation (HGT-SOP-B003 1.1) microarray was utilized to determine the methylation levels of more than 865,000 cytosines followed by guanine residues (CpG) in 32 healthy women. Bisulfite conversions of 500 ng of genomic DNA were performed using the EZ DNA methylation kit (D5002, ZymoResearch, Irvine, CA, USA) according to the instructions and the different incubation settings recommended by the manufacturer. The cycling conditions of 16 cycles were 95 °C for 30 s, 50 °C for 1 h, 4 °C until the purification stage, and then elution in 12 µL. QPCR was used to confirm DNA bisulfite conversion. Subsequently, 4 µL of the eluate containing bisulfite-converted DNA was examined using an Illumina Infinium Methylation EPIC array according to the manufacturer’s instructions. Arrays were scanned on an Illumina iScan Reader, San Diego, CA, USA. Eight array controls were utilized—staining controls, extension controls, hybridization controls, target removal, bisulfite conversion control (1–2), specificity controls (1–2), nonpolymorphic controls, and negative controls. For each CpG site, a beta value is calculated by dividing the methylation signal by the total of the methylated and unmethylated signals. The beta values, which range from 0 (unmethylated) to 1 (fully methylated), denote the level of methylation of a particular CpG site in the sample.

A total of 1 µg RNA per sample was prepared. Following the instructions of the manufacturer, sequencing libraries were created using TruSeq Small RNA Library Preparation Kits from Illumina (San Diego, CA, USA). In brief, 3′ and 5′ ends of short RNA were ligated with 3′ and 5′ adaptors, respectively. Then, SuperScript II Reverse Transcriptase was used to create first-strand cDNA. After PCR amplification, the library was size-selected with 115–160 bp using the BluePippin system. Using a real-time PCR instrument and an Agilent Bioanalyzer 2100 system, the quality of purified libraries was evaluated. On the Illumina NextSeq 500 platform, the qualifying libraries were subsequently sequenced using 75 bp single-end reads produced by Genomics, BioSci and Tech Co. (New Taipei City, Taiwan).

### 2.5. Differential Analysis of Methylated DNA and miRNA

In this study, the analysis of DEM genes between FHBC and non-FHBC was performed using the linear models for microarray (LIMMA) package version 3.62.2, a crucial element of Bioconductor (a statistical genomics open-source software development project based on R) [16,17]. In this study, DNA methylation levels are depicted using beta values ranging from 0 to 1. The M/(M + U + 100) formula was used to compute the methylation level of each CpG. M and U represent methylated and unmethylated intensity, respectively. Differential analysis using LIMMA revealed 1136 of 865,000 CpGs with a corresponding *p* value of <0.05. Additionally, the Bioconductor package was used to perform differential expression analysis of miRNA between the two groups based on the negative binomial distribution (DESeq2). DESeq2 combines methodological breakthroughs with numerous unique features to facilitate a quantitative examination of comparative RNA-seq data by using dispersion and fold-change shrinkage estimators [18]. This analysis revealed a total of 452 of 1730 hsa-miR with a corresponding *p* value of <0.05.

To account for multiple hypothesis testing and to reduce the likelihood of false positives, we applied the Benjamini–Hochberg (BH) procedure to control the false discovery rate (FDR) in both the DEM and DEmiRNA analyses. This method ranks the individual *p* values from smallest to largest and adjusts them to control for the expected proportion of false discoveries among the significant results. For the DEM analysis, FDR correction was applied to the CpG site-level statistical comparisons to identify significantly differentially methylated CpGs while minimizing type I errors. Similarly, for the DEmiRNA analysis, the BH procedure was used to adjust the significance levels of differentially expressed miRNAs, ensuring that the reported miRNAs were statistically significant after multiple testing corrections. Only CpG sites and miRNAs with an FDR-adjusted *p* value (q value) < 0.05 were considered statistically significant, ensuring stringent selection criteria for biologically relevant findings.

### 2.6. Determination of Potential Genes

#### 2.6.1. Phase One: Functional Overlap Between DEM Genes in the Current Study and GEO Database

The GEO database is a worldwide repository of scientific community-donated, high-throughput functional genomics data from microarrays, next-generation sequencing, and other sources [19]. This study utilized the GEO database to identify and validate potential genes in relation to three biopsychosocial factors (i.e., family history, age, and alcohol consumption). DEM gene profiles and clinical data of GSE88883 were downloaded from the GEO database (https://www.ncbi.nlm.nih.gov/geo/query/acc.cgi?acc=GSE88883, accessed on 1 March 2022) to identify and validate the potential biomarkers in normal breast tissue. This data set consists of the breast tissue biopsy specimens of 100 women without cancer, who donated the specimens to the Susan G. Komen Tissue Bank, and includes information on breast cancer risk factors such as age, BMI, parity, and a family history of the disease. The DEM gene samples of this data set were categorized into two groups—FHBC and non-FHBC. In total, 90 of 100 samples were retained.

Another data set from the GEO database was used to identify and validate potential biomarkers in relation to age and alcohol consumption among patients with breast cancer. GSE67919 was downloaded from the GEO database (https://www.ncbi.nlm.nih.gov/geo/query/acc.cgi?acc=GSE67919, accessed on 4 April 2022). The GSE67919 data set includes the data of 324 women with invasive breast cancer. The tissue specimens of 96 women were identified and included in the analysis. This data set also includes data on gender, ethnicity, BMI, alcohol consumption, menopausal status, age at surgery, diagnosis at surgery, and type of surgery. We were specifically interested in analyzing age and alcohol consumption to determine potential biomarkers. We used GEO2R to analyze DEM genes with a *p* value of <0.05. The CpG screening strategy is presented in Figure 2A. After obtaining three profiles of DEM genes from the GEO data set and one profile of DEM genes from the current study, we constructed a Venn diagram using Venny (https://bioinfogp.cnb.csic.es/tools/venny/, accessed on 10 April 2022) to show the overlap of DEM genes.

#### 2.6.2. Phase Two: Functional Overlap Between Phase One’s DEM Genes and DEmiRNA to Reveal Potential Genes

We applied a gene screening strategy involving the overlap of the DEM genes from phase one and DEmiRNA. The data set from the current investigation was used to transform the gene IDs of DEM genes into the appropriate gene symbols. Uncharacterized genes were removed, and 95 potential genes from the list of DEM genes remained. Moreover, we used the Target Scan Human database (https://www.targetscan.org/vert_80/, accessed on 10 February 2022) to predict DEmiRNA target genes. We inputted 10 hsa-miRNAs that have high predictive scores. In total, 5151 genes were found in this database. A Venn diagram was used to show the overlap of potential genes.

### 2.7. Prognostic Value of Potential Genes in Breast Cancer

#### 2.7.1. Gene Expression Profiles: UALCAN Analysis

UALCAN is an online resource for studying TCGA gene expression data. Microarray sequencing data from various sub-statuses are used in this website to determine expression patterns related to stage, grade, ethnicity, and other sub-statuses [20]. We verified the expression levels of two previously identified potential genes in 1097 primary breast tissue samples and 114 normal breast tissue samples (http://ualcan.path.uab.edu/analysis.html, accessed on 6 April 2022). To verify the importance of these two potential genes in healthy and breast cancer tissues, we also performed an independent *t*-test. A difference with a *p* value of <0.05 was considered significant. We also investigated the mRNA expression patterns of tumor stage markers, and the breast cancer patients’ race. A trend of larger statistically significant differences in more advanced tumors when compared by race was observed.

#### 2.7.2. Survival Analysis of Two Potential Genes

In this study, survival analysis was performed using the MethSurv database (https://biit.cs.ut.ee/methsurv/, accessed on 11 July 2022), which provided the methylation levels of the two potential genes. MethSurv enables survival analysis using the R package to examine the methylation level of a CpG located near or next to a query gene. For a query gene, the optimal biomarkers for each cancer are provided for browsing, and cluster analysis is performed to correlate methylation patterns with clinical characteristics [21]. We examined the methylation pattern of a single CpG in the two potential genes in patients with breast cancer using this database to ascertain expression and prognostic trends. In this study, DNA methylation levels are represented using beta values (ranging from 0 to 1). The methylation levels of all CpGs were measured using the M/(M + U + 100) formula, where M and U represent methylated and unmethylated values, respectively. The hazard ratio, the log likelihood ratio (LR) test, *p* value, and information on the query gene are presented on Kaplan–Meier graphs. We also explored breast cancer survival using a cohort study from the TCGA database. We used Cox regression analysis to explore the associations of breast cancer survival with age, cancer stage, T stage, and treatment.

#### 2.7.3. cBioPortal and GeneMANIA Analysis of Two Potential Genes

A multidimensional cancer genome data set is accessible through the cBio Cancer Genomics Portal (http://cbioportal.org, accessed on 2 August 2022). The cBioPortal includes the results of more than 200 cancer genomics studies, including all of the data from TCGA [22]. The portal transforms molecular profiling information from cancerous tissues and cell lines into easily comprehensible data on genetic, epigenetic, gene expression, and proteomic events [23]. The TCGA samples of breast cancer patients were examined in this study using the cBioPortal to identify two potential gene alterations. Through this database, we explored alterations in GNPDA1 and SLC25A16 in various invasive breast cancer tissues. OncoPrint was used to obtain a summary of genetic alterations in the two potential biomarkers. GeneMANIA is an analytical web tool for generating assumptions based on gene activities (www.genemania.org, accessed on 8 August 2022). GeneMANIA can perform a search, provide a set of genes whose functions are comparable to those of the gene target, and create an interactive network to show how the target gene and the data set are related [24]. Using GeneMANIA, we created a gene–gene interaction network connected to our potential genes in this study.

#### 2.7.4. Functional Enrichment Analysis of Potential Genes

In gene ontology, the concept of systematically connecting a group of genes with a functional biological term has been established [25]. We utilized enrichment analysis using the Enrichr database (https://maayanlab.cloud/Enrichr/, accessed on 10 August 2022) to provide the enriched categories of genes for biological processes, cellular components, and molecular functions through gene ontology and the Kyoto Encyclopedia of Genes and Genomes (KEGG). Enrichr is a comprehensive database of gene sets that have been carefully selected, and it includes a search engine that accumulates biological knowledge [26]. Additionally, this study employed an integrative analysis utilizing a number of databases, including cBioPortal and MetaCore. We utilized the cBioPortal online tool to investigate relationships between the GNPDA1 and SLC25A16 genes. We selected co-expressed genes with a correlation value of ≥0.2 with the potential biomarkers. Moreover, we used the MetaCore platform (https://portal.genego.com, accessed on 23 August 2022) to more fully identify the signaling pathways and biological functions of potential genes in breast cancer.

#### 2.7.5. Mapping Drug Targets for Breast Cancer Using Drug Database

To evaluate the potential drug targets of two potential biomarkers (GNPDA1 and SLC25A16), we utilized the STRING database to identify functional interactions related to protein expression by integrating predicted protein–protein association data. The STRING database version 12.0 (https://string-db.org/, accessed on 11 October 2023) was used to identify protein–protein interactions (PPIs) with these encoded by two potential genes. We defined the genes that encode these proteins as “drug target genes”. Several protein products derived from each of the two potential genes were studied as potential targets for approved drugs, either as breast cancer therapeutics or for other therapeutic purposes.

The DrugBank database (https://go.drugbank.com/, accessed on 14 October 2023) and Therapeutic target database (https://db.idrblab.net/ttd/search/ttd/target, accessed on 14 October 2023) were utilized to evaluate potential drug targets according to several criteria, including drugs with pharmacological activities, efficacy in humans, and annotations of ‘approved’, ‘clinical trial’ or ‘experimental drugs’. We reviewed ClinicalTrials.gov (https://clinicaltrials.gov, accessed on 15 October 2023) for any clinical trials related to breast cancer or other diseases based on the identified drugs.

## 3. Results

### 3.1. Population Demographics

Table 1 provides a summary of participant characteristics, including biological, psychological, and social characteristics of the FHBC and non-FHBC groups. Regarding biological factors, breast cancer risk demonstrated a significant correlation among women relating to age (*p* < 0.001), age at menarche (*p* = 0.040), age at first live birth (*p* < 0.001), mother’s breast cancer history (*p* < 0.001), sister’s breast cancer history (*p* < 0.001), and breastfeeding practices (*p* < 0.001). Furthermore, psychological and behavioral characteristics, including alcohol use (*p* = 0.011), smoking (*p* = 0.008), and exercise habits (*p* = 0.011), exhibited significant differences across the groups. Additionally, social factors such as job experience (*p* < 0.001) and living alone (*p* = 0.020) exhibited significant differences.

### 3.2. Identification of DEM and DEmiRNA

Figure 2 presents significant molecules discovered by independent analyses of DEM genes and DEmiRNA between FHBC and non-FHBC using LIMMA and DESeq2, respectively. A volcano plot was used to display significant methylated CpGs (Figure 2A). The top 10 CpGs are listed in Appendix A. According to this analysis, cg23786701 was the most significant DEM gene (*p* = 5.98 × 10^−14^). A volcano plot was also used to display significant hsa-miRNAs (Figure 2B). The top 10 hsa-miRNAs are listed in Appendix A, and hsa-let-7p-5p was the most significant hsa-miRNA (*p* = 4.15 × 10^−46^). Significant CpG sites and hsa-miRNAs are indicated in red (downregulated) and green (upregulated).

### 3.3. Identification of Potential Genes (GEO Database and Current Study)

To reveal potential genes in this study, we applied two main screening strategies (Figure 3). Phase one involved investigating the functional overlap between DEM genes from the GEO database in relation to three biopsychosocial factors (family history, age, and alcohol consumption) and genes from the current study. A GEO2R analysis from the GEO database (GSE88883 and GSE67919) was used to produce 390,293 CpGs for family history (FH) of cancer, 375,421 CpGs for factor age, and 378,906 CpGs for alcohol consumption.

The LIMMA package was used to generate 865,919 DEM genes from the current study. In total, 12,907 of 390,293 CpGs relating to the FH, 49,237 of 375,421 CpGs relating to age, 22,476 of 378,906 CpGs relating to alcohol consumption, and 1136 of 865,919 CpGs from the current study were selected with *p* values of <0.05 (Figure 3A). Furthermore, using a Venny diagram, we identified six areas where DEM genes intersect (Figure 3B). Different colors represent different data sets, and the functional overlap regions show important CpGs. We found 137 CpGs of interest in total with 95 known genes. The final potential genes have been revealed in phase two. Phase two involved investigating the functional overlap between 95 potential genes from phase one and 5151 potential genes from DEmiRNA (Figure 3C). Phase two resulted in the final potential genes. A final list of potential genes is provided in Table 2. Finally, 2 of 11 genes of interest in the promotor sites (i.e., GNPDA1 and SLC25A16) were identified as potential genes.

The majority of our significant findings remained statistically significant after applying FDR correction, especially among the top 10 CpGs of differentially expressed methylation sites (Appendix A) and the hsa-miRs of differentially expressed miRNAs (Appendix A). This demonstrates that our results are robust and not merely due to random chance. While GNPDA1 and SLC25A16 CpG sites exhibited FDR values greater than 0.05, indicating that the changes in methylation did not reach statistical significance after correction the corresponding hsa-miRs targeting these genes maintained FDR values below 0.05 (Table 2). This indicates that, while the direct methylation alterations in these genes may lack statistical significance post-correction, their regulatory miRNAs exhibit significant differential expression. This association emphasizes the potential role for epigenetic regulation in these genes and indicates that miRNA-mediated post-transcriptional regulation may remain a crucial mechanism in familial breast cancer risk. The persistence of significant findings following FDR correction demonstrates the robustness and reliability of our results, diminishing the probability of false positives while reinforcing biologically relevant targets.

### 3.4. The Role of Potential Genes in Breast Cancer Prognosis (TCGA Database)

Using the UALCAN data analysis portal, we analyzed breast cancer patients from the TCGA database to investigate mRNA expression patterns of the two potential genes and clinical factors including tumor stage and the patient’s race (Figure 4 and Appendix A). Statistically significant differences (*p* < 0.05) were observed in the expression of GNPDA1 and SLC25A16 between the normal and tumor tissues (Figure 4A,B, left). On the basis of tumor stage indicators, we found that in more advanced tumors, GNPDA1 and SLC25A16 mRNA expression tended to have more statistical significance (Figure 4A,B, middle). Moreover, we observed that the expression levels of GNPDA1 and SLC25A16 were statistically significant in relation to the racial background of the patient (Appendix A). The Methsurv web tool was used to perform prognostic survival analysis of methylated DNA from the TCGA database. Kaplan–Meier survival analysis of methylation related to breast cancer was carried out using the R package. We demonstrated a significant correlation between poor prognosis and the hypomethylation of GNPDA1. By contrast, the hypermethylation of SLC25A16 was significantly associated with poor prognosis (Figure 4A,B, right).

CpG island methylation by DNA methyltransferases is a reversible process, through which transcription factors either promote or inhibit cell development. Using the Methsurv web tool, a heat map demonstrated the significance scores of DNA methylation which predict expression levels of GNPDA1 and SLC25A16 in patients with breast cancer (Appendix A). According to the methylation levels, the cg00145118 of GNPDA1 and cg00203461 and the cg10590909 and cg16214034 of SLC25A16 had the highest prognostic value in patients with breast cancer (LR test *p* < 0.05). Furthermore, Cox regression analysis was used to examine the correlations of breast cancer survival with clinical data from TCGA, including age, tumor stage, T stage, and treatment (Appendix A). We found that breast cancer survival was correlated with age and cancer stage (especially stages III, IV, and X; *p* < 0.05).

### 3.5. Genetic Alteration, Gene Interaction, and Neighbor Gene Network of Two Potential Genes in Patients with Breast Cancer

Using the cBioPortal online tool (https://bit.ly/3OVsxFd, accessed on 2 December 2022), genetic alterations, gene networks, and the gene interactions of the two potential genes in patients with breast cancer were analyzed. We examined 994 data samples of breast invasive carcinoma from the TCGA and PanCancer Atlas databases. The result shows that breast invasive carcinoma (NOS) appears to have a much higher frequency of GNPDA1 and SLC25A16 (15,49%) than the other subtypes of breast carcinoma (Appendix A). The distribution of genetic alterations of the two potential genes was 110 (11%) of queried samples (GNPDA1: 6% and SLC25A16: 5%) as shown in Appendix A. Further, an interaction analysis of the potential genes was performed using GeneMANIA to clarify the correlations among physical interaction, co-expression, predicted, colocalization, genetic interactions, pathways, and shared protein domain (Appendix A). It was determined that physical interactions (77.64%) were the most common interaction between the two potential genes and several gene products/proteins included GNPDA2, AMDHD2, SDHB, ALDH2, SUCLG1, DPH5, CS, LRRK2, ACCS, TBCE, FH, CDC34, and TPP1.

### 3.6. Gene Ontology, KEGG, and Genome Pathway Exploration of Two Potential Genes in Breast Cancer

The Enrich database was used to analyze gene ontologies and KEGG pathways to determine the biological function, pathway, diseases, cell lines, and their neighboring genes of two potential genes. The top enrichment terms of the input gene set provide insight into the gene set’s characteristics [27].

According to the gene ontology assessment of the co-expressed genes, the biological processes for GNPDA1 mostly involve the regulation of cellular amino acid, amine metabolic processes, and mitochondrial translational elongation (Appendix A). Molecular functions that were highly regulated included RNA binding, methyl-CpG binding, and endopeptidase activity (Appendix A). Cellular components that were mostly regulated included ficolin-1-rich granule lumens and mitochondrial and organelle inner membranes (Appendix A). KEGG pathway analysis revealed proteasome, spinocerebellar ataxia, and prion disease (Appendix A). Vesicle organization, Aspartate family amino acid catabolic processes, and ubiquitin-dependent protein-catabolic processes were included in the biological process of SLC25A16 (Appendix A). Molecular functions included RNA polymerase II transcription, regulatory region sequence-specific DNA binding, miRNA binding, and ATPase regular activity (Appendix A). Cellular components include the apicolateral plasma membrane, RISC-loading complex, and extrinsic endosome membrane (Appendix A). KEGG pathway analysis revealed the enrichment of herpes simplex virus 1 infection and the ABC transporter, which regulate cancer-related signaling pathways (Appendix A).

Using MetaCore, we examined the co-expressed genes for GNPDA1 and SLC25A16 from the cBioPortal and TCGA breast cancer data sets. The MetaCore tool analyzes gene inputs and identifies pathways that are networked to stimulate biological processes. As a result of the gene list as input in MetaCore analysis, we discovered some interesting results regarding GNPDA1 and SLC25A16 in the development of breast cancer. First, GNPDA1 was strongly associated with “DNA damage_ATM activation by DNA damage (*p* = 4.282 × 10^−06^)”, “BRCA1 and BRCA2 in breast cancer (*p* = 6.950 × 10^−04^)”, and “DNA damage_ATM/ATR regulation of G2/M checkpoint: nuclear signaling (*p* = 8.093 × 10^−04^)” (Figure 5A). This indicates that GNPDA1 is linked to several genes such as BRCA1 and RAD50 and is associated with DNA damage and breast cancer (Appendix A). BRCA1 and BRCA2 are linked to multiple genes that affect the suppression of homologous recombinational repair (HDR) and lead to breast cancer development by promoting cell proliferation, mitotic aberrations, and genomic instability, as shown in the signaling pathway in Figure 5B. Second, genes co-expressed with SLC25A16 are involved in “Pro-oncogenic action of androgen receptor (*p* = 2.114 × 10^−04^)”, “development positive regulation of WNT/Beta catenin signaling in the nucleus (*p* = 2.957 × 10^−04^)”, and “transport clathrin-coated vesicle cycle (*p* = 3.380 × 10^−04^)” (Appendix A). These findings suggest that the differential expression of genes encoding androstenedione and testosterone leads to breast cancer cell progression through cell proliferation and cell migration and influences the metastasis mechanism in breast cancer (Appendix A).

### 3.7. Discovery of Breast Cancer Drug Targets

In order to identify therapeutic candidates for breast cancer, we aligned genes with relevant drugs within the DrugBank and therapeutic target databases. We obtained 28 gene pairs from the STRING database curated as part of the PPI network for SLC25A16 (Appendix A). It was found in DrugBank that two target genes (PDCD1, CD274) overlapped with six drugs (Nivolumab, Pembrolizumab, Cemiplimab, atezolizumab, Durvalumab, and Avelumab). All of which are either approved in clinical trials or are experimental drugs for human diseases. As seen in Figure 6, six drugs are currently under clinical investigation for the treatment of breast cancer. It is important to note that two target genes with six drugs may have the potential to be repositioned for the treatment of breast cancer. In addition, we obtained 100 gene pairs curated for PPI networking for GNPDA1 from STRING database (Appendix A). According to the Therapeutic target database, our study shows that four target genes (G6PD, GPI, MPI, and PGD) are being crossed with five drugs (Prasterone, Lutetium Lu-177 Vipivotide Tetraxetan, Bevirimat, Praziquantel, and Laropiprant). The drugs are either approved for use in humans, are in clinical trials, or are experimental drugs that are being investigated for use in breast cancer treatments. However, not all the drug target genes that we identified are involved in pharmacological activities, which may mean that they are not likely to be discovered (undruggable).

## 4. Discussion

This study evaluated the epigenetic mechanisms of biopsychosocial factors in Taiwanese women with a family history of breast cancer through an integrated analysis of DNA methylation and miRNA. The crucial personal information and biopsychosocial factors linked to breast cancer risk mostly encompass biological characteristics such as age, age at menarche, age at first live birth, a mother who has had breast cancer, a sister who has had breast cancer, and breastfeeding practice. Breast cancer development is influenced by various factors, including age [28,29], a family history of cancer [30], reproductive variables [31], lifestyle choices, and psychosocial factors [1,4,5,32]. One study in the United Kingdom [33] revealed that the incidence of breast cancer increases with age until menopause. The risk of breast cancer is higher among women with a family history of cancer than among those without a family history of cancer. The incidence of breast cancer is higher among women who have a first-degree relative with breast cancer than among women without a family history of cancer [32].

Psychosocial factors, including lifestyle factors, are considered risk factors for breast cancer. Alcohol consumption, smoking, and exercise are risk factors for breast cancer in this study. Consumption of as little as 10–15 g of alcohol per day can lead to breast cancer. Additionally, although the data are less conclusive, they demonstrate that alcohol use is linked to potential early risk markers such as benign breast disease and increased breast density [34]. In relation to the smoking variable, recent research has connected smoking and exposure to secondhand smoke to an increased risk of breast cancer in women. Carreras et al. found that smoking and secondhand smoke exposure resulted in 2.6% and 1.0% of breast cancer deaths and disability-adjusted life years lost, respectively [35]. Furthermore, another previous study provides strong evidence that exercise, vigorous activity, and a decreased amount of sedentary time are associated with a lower risk of breast cancer. Job experience and living alone are significant indicators of breast cancer incidence for social factors. A study by Vona-Davis and Rose indicated that a low socioeconomic status may be causally associated with an increased vulnerability to breast cancer [36]. Moreover, a crucial factor that may substantially influence the disparities in breast cancer outcomes is the widespread imbalance in socioeconomic position, typically marked by low family income, insufficient level of education, and inadequate access to healthcare insurance. The presence of these deficiencies may result in the underutilization of screening mammography programs [36,37].

Researchers have investigated the relationship between breast cancer risk factors and DNA methylation, particularly in the blood [9]. In the current study, we utilized the GEO database to identify and validate CpGs of interest in relation to three biopsychosocial factors (a family history of cancer, age, and alcohol consumption). By examining the overlap between interesting CpGs and miRNA, we found that GNPDA1 and SLC25A16 are potential genes. Glucosamine-6-phosphate isomerase 1, also known as GNPDA1, is a family of glucosamine-6-phosphate deaminases that increase the raw materials available for glycolysis by connecting the glycolytic pathway with the hexosamine system and then converting glucosamine into fructose 6-phosphate [38]. Glucosamine-6-phosphate deaminases are involved in metabolic pathway reprogramming, which is a defining feature of pathogenic alterations by cancer cells. For tumor cells to evolve into more aggressive phenotypes, several genes of the glucosamine-6-phosphate deaminases family that directly regulate crucial metabolic processes, such as glycolysis, lipogenesis, and nucleotide synthesis, exhibit abnormal expression [39]. A few studies have shown that GNPDA1 influences the development of several cancers, such as hepatocellular carcinoma, gastrointestinal cancer, ovarian cancer, and colorectal cancer [38,40,41,42,43]. While specific studies directly linking GNPDA1 expression to DNA methylation and miRNA are limited, it is plausible that methylation of the GNPDA1 promoter region could reduce its expression. This mechanism has been observed in other genes, where promoter methylation leads to transcriptional repression [44].

SLC25A16, a member of solute carrier (SLC) family 25, encodes a protein with three mitochondrial carrier protein regions that are tandemly duplicated. Various types of genodermatoses have been linked to mutations in other SLC family members [45]. For metabolic reprogramming in cancer, a SLC1A5 variant functions as a mitochondrial glutamine transporter [46]. SLC25A16 is linked to diseases such as non-syndromic congenital nail disorders and isolated nail anomalies [45]. There is also limited information on the direct regulation of SLC25A16 by DNA methylation. However, given that DNA methylation is a common mechanism for regulating gene expression and miRNAs play a significant role in regulating genes involved in metabolic processes. It is plausible that SLC25A16 could be subject to such regulation. Understanding these epigenetic mechanisms provides insight into potential regulatory pathways. Further research is needed to elucidate the specific interactions affecting GNPDA1 and SLC25A16 expression.

Johnson et al. [1] determined the molecular alterations connected to risk factors for cancer in normal tissues and showed that age-related DNA methylation alterations were predominantly characterized by increased methylation in enhancers in mammary epithelial cells (*p* = 7.1 × 10^−20^). By contrast, a family history of cancer was associated with promoter hypermethylation [47]. Several CpG sites were consistently visible in the ESR1 amplicon. The mean methylation levels were higher in girls with a family history of breast cancer than in those without a family history of breast cancer [8]. Studying breast cancer cases is crucial to finding methylation markers that can be used as early detection biomarkers in blood, especially in family cases with early-onset tumors [8,48]. Zhou et al. [49] found that when controlling for alcohol consumption, problematic alcohol use was associated with an increased risk of breast cancer (OR: 1.76, 95% CI: 1.04–2.99). Additionally, epigenetic MR analysis in their study revealed four CpG sites whose genetically predicted epigenetic alterations were linked to a higher incidence risk of breast cancer [49].

mirRNA has been shown to regulate proliferation, differentiation, and apoptosis, particularly so in cancer development and progression [50]. The present study notes two hsa-miRNAs that are related to the two potential genes (GNPDA1 was regulated by hsa-miR-23a-3p, and SLC25A16 was regulated by hsa-miR-425-5p). Studies have indicated that these two miRNAs have important roles in breast cancer development. Firstly, hsa-miR-23a-3p is one of the alias symbols for miR-23a gene. Diseases associated with miR-23a include hepatocellular carcinoma [51,52], ovarian serous carcinoma [53], and breast cancer [54,55]. A significant increase in miR-23a expression has also been observed in breast cancer patients with lymph node metastases, as compared to those with no lymph node metastases or normal tissue. In addition, a high correlation coefficient for the expression of the individual members of the miR-23a gene indicates that cluster co-expression occurs in breast cancer [54]. Secondly, in a study conducted by Zheng et al. [56], hsa-miR-425-5p was identified as one of ten miRNAs that showed abnormal expression in breast cancer tissues when compared to normal tissues. Furthermore, hsa-miR-425-5p was implicated in the link between necroptosis and cancer metastasis, resulting in decreased necroptosis, which reduced the inflammatory response and acute liver damage [56,57]. However, there is currently no direct experimental evidence linking these specific miRNAs to the regulation of GNPDA1 or SLC25A16. Hence, future studies could adopt an experimental approach to explore these potential interactions and clarify the regulatory mechanisms involving these miRNAs and the genes of interest.

According to our bioinformatics analysis results, GNPDA1 and SLC25A16 have much prognostic value—they play key roles in genetic alterations and signaling pathways in breast cancer. Using the UALCAN database, we investigated mRNA expression levels, connections with clinical pathological indicators, and the prognostic value of the two potential genes in patients with breast cancer. GNPDA1 expression was enhanced in primary tumor cells compared to normal cells. Additionally, we found that more advanced tumor stages tended to have significantly higher GNPDA1 mRNA expression based on the clinical and pathological features of the tumor stage. Meanwhile, compared to normal tissues, the SLC25A16 gene was considerably underexpressed in human breast cancer tissues. It was correlated with the mRNA expression of individual cancer stages which were significantly lower in more advanced tumor stages. The identification of abnormal gene expression, which is induced by oncogenic activity, is important for the early detection of malignancies [58]. These findings are important because alterations in the expression or structure of proteins can contribute to uncontrolled proliferation [58,59]. Based on our findings, the expression levels of GNPDA1 and SLC25A16 were significantly higher and lower, respectively, among three races of patients (Caucasian, African American, and Asian) than in normal tissue. This evidence demonstrated that these two potential genes may indicate the patient’s markers according to different heredity backgrounds. According to previous studies, invasive breast cancer is more aggressive in Black women compared to white women from a pathological and biological standpoint [60,61]. Sweeney et al. [62] described intrinsic subtypes in breast cancer based on a gene expression assay in a U.S. population-based study, as well as in a study population representing a variety of ethnicities and racial groups. In addition, gene expression-based subtyping has also been demonstrated to support the notion of molecular heterogeneity within subtypes and to indicate distinct intrinsic subtype profiles by age and race [62].

Despite the current study utilizing blood samples, we emphasized that circulating DNA methylation and miRNA patterns in blood plasma can serve as indicators of systemic changes and may reflect underlying processes occurring in various tissues [63]. Numerous investigations have validated a significant correlation between genetic and epigenetic modifications identified in plasma ctDNA (circulating tumor DNA) and circulating miRNA that correlate with those in tumor tissue [63,64,65]. The existence and dynamics of ctDNA and circulating miRNA could transform cancer screening, diagnosis, and therapy through a noninvasive method [63,65]. While DNA methylation and miRNA expression patterns are often tissue-specific and play critical roles in gene regulation, the primary goal of our study was not to fully elucidate the intricate tissue-specific regulatory mechanisms but rather to explore the feasibility of using blood-based biomarkers for clinical applications. This can provide valuable insights into risk assessment and early detection.

Our comprehensive study revealed that GNPDA1 and SLC25A16 have prognostic value in patients with breast cancer. The hypomethylation of GNPDA1 is associated with poor survival. Meanwhile, SLC25A16 hypermethylation is associated with poor survival in patients with breast cancer. According to the TCGA database, age, cancer stage, T stage, and treatment were associated with the tumor progression stage. According to the DNA methylation analysis in the current study, GNPDA1 and SLC25A16 in a single CpG serve as prognostic biomarkers of breast cancer. DNA methylation levels have prognostic value for cg00145118 from GNPDA1 and cg00203461, cg10590909, and cg16214034 from SLC25A16. Therefore, these genes have prognostic value, especially among healthy women with a family history of cancer. Several studies have revealed that GNPDA1 is one of the nine-gene amino acid metabolism-related risk signatures that are constructed to indicate the overall survival of hepatocellular carcinoma patients [38,42]. Moreover, GNPDA1 was associated with protein upregulation and was noted to play a role in the biological processes underlying anti-cancer activity against colorectal cancer [66] and gastrointestinal cancer [41]. Furthermore, SLC25A16 is linked to illnesses such as non-syndromic congenital nail disorders and isolated nail anomalies [45].

The distribution of genetic changes for specific candidate genes revealed that GNPDA1 and SLC25A16 mRNA levels were elevated in approximately 5% of breast cancer samples. Moreover, we investigated gene interactions and the network of the two potential genes. The results of GeneMANIA showed that the most common interactions between the two potential genes and several other genes were physical interactions (77.64%). GNPDA2, AMDHD2, SDHB, ALDH2, SUCLG1, DPH5, CS, LRRK2, ACCS, TBCE, FH, CDC34, and TPP1 are 12 genes involved in the physical interactions of GNPDA1 and SLC25A16, and these 12 genes were identified in the 26S proteasome [67], chemokine [68], proteomes [69], and mitochondria interactions [70]. 26S proteasomes contribute 4.41% of the physical interactions that influence disease and malignancies. Due to the high demands of malignant cells, the ubiquitin–proteasome system can become unregulated in diseases such as cancer. Proteins are produced and degraded at an excessive pace by rapidly dividing malignant cells, resulting in damaging effects such as cachexia and fatigue [71].

Genes co-expressed with the two candidate biomarkers were analyzed using gene ontology from the Enrichr database. The biological processes of GNPDA1 included the regulation of cellular amine and amino metabolic processes [72] and mitochondrial translational elongation [73]. Amino acids and mitochondria play a significant role in the biological activities that influence breast cancer development. Amino acid metabolism is altered in patients with breast cancer, and amino acid transporters influence the development and growth of tumors. One of the most important nutrients for treating breast cancer is glutamine, and amino acid transporters and glutamine metabolism are closely connected [72]. In the present study, the molecular functions of GNPDA1 included the regulation of RNA binding [74,75], methyl-CpG binding [76], and endopeptidase activity. The regulation of RNA and methyl-CpG binding were the main focus of this study, which aimed to reveal potential biomarkers. The discovery that RNA and methyl-CpG binding are associated with the potential genes emphasizes that GNPDA1 is a potential biomarker of breast cancer development. In addition, fcolin-1-rich granule lumen, mitochondrial inner membrane [58], and organelle inner membrane were mostly regulated in the cellular components. Whereas, proteasome [67,69,71], spinocerebellar ataxia, and prion disease [70] were mostly influenced in KEGG.

The biological activities of SLC25A16 included vesicle organization [77,78], catabolic processes of the aspartate family of amino acids [79], and catabolic processes of ubiquitin-dependent proteins through the multivesicular body sorting route [80]. Alterations in the following molecular functions were detected—ATPase activity [80], miRNA binding [79], and sequence-specific DNA binding to the RNA polymerase II transcription regulatory region [77,78]. Cellular components were affected by the apicolateral plasma membrane, RISC-loading complex, and extrinsic endosome membrane. KEGG pathway analysis revealed the enrichment of the pathways for herpes simplex virus 1 infection [81,82] and the pathways involving ABC transporter [77].

The most interesting finding of the MetaCore analysis was the significant correlation between GNPDA1 and BRCA1 and BRCA2 in breast cancer cells. In this pathway, BRCA1 was correlated with the mitogenic action of ESR1 in breast cancer cells [78,79]. BRCA1 was associated with several genes that induce the inhibition of G2/M transition [80,83], centrosome amplification [84,85,86,87], paired centriole [88,89], the inhibition of HDR [90], and the inhibition of transcription-couple repair. BRCA2 was associated with several genes that influence the inhibition of HDR and mechanisms that stimulate cell proliferation, mitotic abnormalities, and genomic instability in breast cancer cells. MetaCore analysis also revealed that pro-oncogenic actions of the androgen receptor were associated with breast cancer [91]. This pathway was also significantly correlated with SLC25A16 in breast cancer development. The mechanisms that affect the androgen receptor in the signaling pathway were the anti-apoptotic and mitogonic actions of ErbB2 [92,93,94], the nuclear translocation of betacatenin [95,96,97,98], and androstenedione and testosterone biosynthesis [99,100,101,102]. The pro-oncogenic actions of androgen receptor-induced cell proliferation, cell migration, the inhibition of cell–cell adhesion, and cell growth, which promoted breast cancer metastasis.

Furthermore, our study also examined six drugs which are currently in clinical trials for breast cancer, including nivolumab, pembrolizumab, cemiplimab, atezolizumab, durvalumab, and avelumab. Nivolumab (NCT03815890), pembrolizumab (NCT05382286), atezolizumab (NCT05180006), and durvalumab (NCT03794596) are currently in phase 3 clinical trials. Whereas cemiplimab (NCT05429866) and avelumab (NCT03794596) are currently in phase 2 clinical trials for breast cancer treatment. Patients with triple-negative breast cancer (TNBC) are more likely to present with advanced disease, have a higher incidence of metastasis and recurrence, and have an extremely poor prognosis [103]. An increase in PDCD1 and CD274 expression was correlated with better overall survival (OS) and disease-specific survival (DSS) in TNBC and breast-invasive carcinoma [104,105]. In this study, PDCD1 and CD274 represent the drug target genes. PDCD1 is targetable by pembrolizumeb, while CD274 is targetable by three drugs (azetolizumab, durvalumab, and avelumab). Interestingly, these four drugs that we identified here have satisfactory safety profiles and may actually promote the treatment of TNBC [106,107,108,109].

Several limitations have been identified in this study. A primary weakness of this study is the relatively limited sample size utilized for DNA methylation and miRNA expression analysis. DNA methylation profiling was performed on 32 subjects, whereas miRNA expression analysis involved just 16 participants. The restricted sample size may diminish the statistical strength of our findings and constrain the generalizability of the results to larger groups. Moreover, inter-individual heterogeneity in DNA methylation and miRNA expression may affect the observed patterns, necessitating a larger cohort to corroborate our findings. Subsequent research using larger sample sizes and independent replication cohorts would enhance the dependability of the findings and offer more thorough insights into the molecular mechanisms driving the observed alterations. Although multiple online databases were used to discover important prognostic values of the two potential genes in breast cancer development, the method is unable to indicate which cell types are affected by epigenetic changes. It is also important to note that this study lacks clinical validation and genetic mutation screening among participants. Future prospective studies are therefore needed to investigate the pattern of methylation and miRNA expression changes in GNPDA1 and SLC25A26 in a cohort population with breast cancer and incorporate comprehensive genetic screening to further elucidate the relationship between hereditary mutations, family history, and molecular alterations. The functional link between both mechanisms of these two genes particularly in relation to biopsychosocial factors and the genes targeted by the drugs still requires further investigation to ascertain the role of the genes.

## 5. Conclusions

In Taiwanese women with FHBC, our study identified GNPDA1 and SLC25A16 as candidate genes through an integrated analysis of DNA methylation and miRNA. Age, a family history of cancer, and alcohol consumption were the biopsychosocial factors associated with the two potential genes. GNPDA1 overexpressed hypomethylation and SLC25A16 underexpressed hypermethylation in breast cancer cells were associated with poor prognosis in patients with breast cancer. GNPDA1 and SLC25A16 play important roles in the BRCA1, BRCA2, and pro-oncogenic actions of the androgen receptor in breast cancer development. These genes are potential biomarkers for the early detection of breast cancer in women, especially those with a family history of breast cancer. Further, we found that GNPDA1 and SLC25A16 were enriched in known targets of approved cancer drugs as potential genes associated with breast cancer. Using this approach, candidate drugs can be narrowed down before clinical trials, providing new avenues for developing breast cancer drugs, enhancing the discovery process for new drugs, and providing potential gene targets and potential drugs for the repurposing of existing drugs to treat breast cancer.

## Figures and Tables

**Figure 1 jpm-15-00134-f001:**
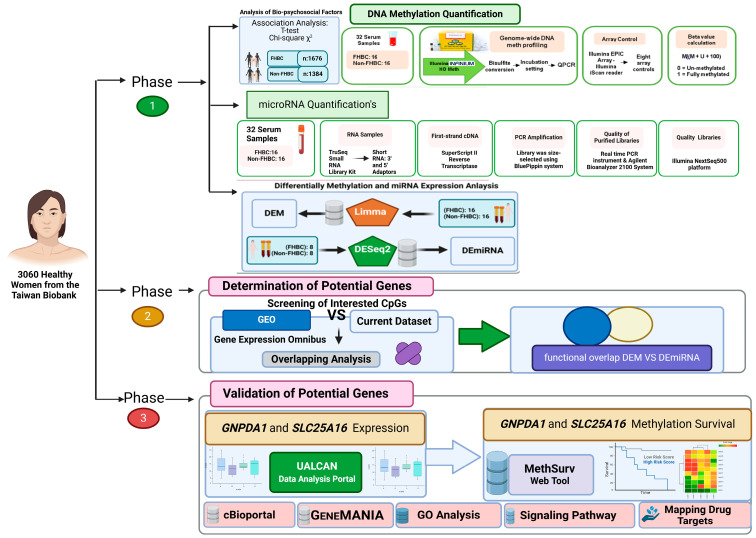
Phases of analytical approach and research design. FHBC, family history of breast cancer; QPCR, quantitative polymerase chain reaction; cDNA, complementary DNA; DEM, differentially methylated; DEmiRNA, differetially expressed microRNA LIMMA, linear models of microarray; DESeq2, differential gene expression analysis of RNA-seq2; GEO, Gene Expression Omnibus; UALCAN, University of ALabama at Birmingham CANcer; MethSurv, metylation survival.

**Figure 2 jpm-15-00134-f002:**
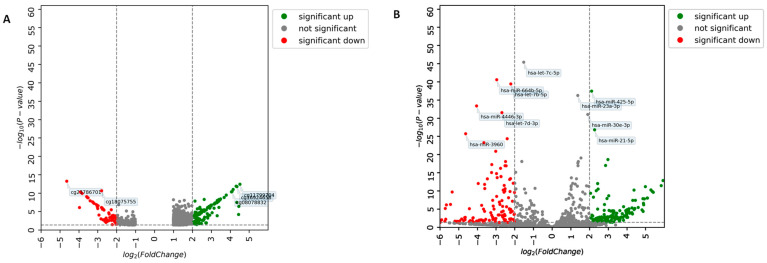
Differential analysis of methylated DNA and miRNA. (**A**) A volcano plot of DEM genes in the FHBC and non-FHBC groups. Significant CpGs in a volcano graphic with logFC ≥ 1.0 (*p* < 0.05). (**B**) A volcano plot of DEmiRNA with logFC ≥ 2.0 (*p* < 0.05).

**Figure 3 jpm-15-00134-f003:**
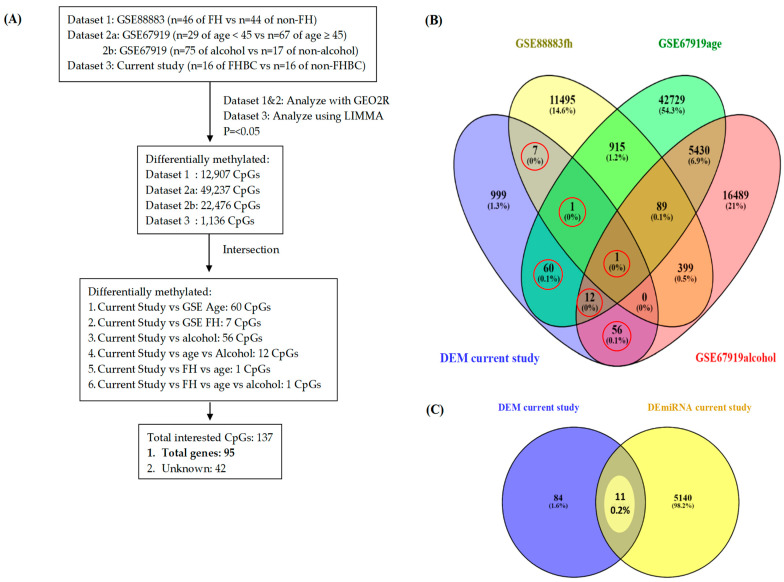
**The identification and validation of potential genes**. (**A**) Screening strategy for CpGs of interest from DEM genes of three data sets. GSE88883 (Data set 1), GSE67919 (Data sets 2a and 2b), current study (Data set 3). (**B**) Different colored areas represent different data sets. The crossed areas correspond to the common differentially methylated. Data sets from the GEO database were analyzed with GEO2R, and the current study data set was analyzed using LIMMA (*p* < 0.05). (**C**) Functional overlap between DEM genes of interest and DEmiRNA of current study. (FH, family history of cancer.)

**Figure 4 jpm-15-00134-f004:**
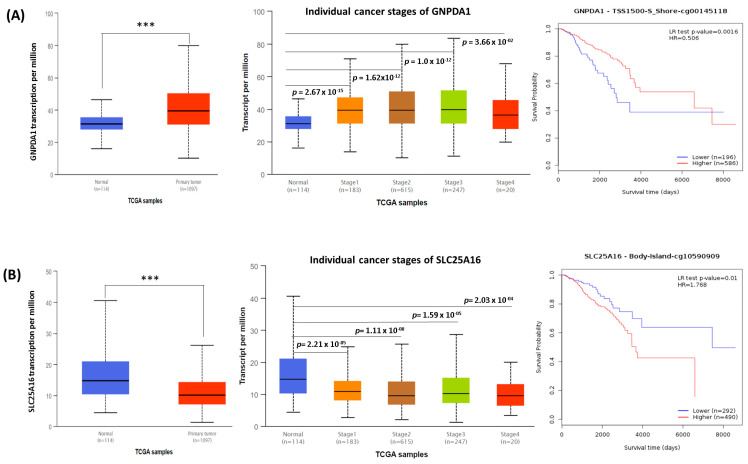
The expression and methylation level analyses of the potential genes in breast cancer (TCGA database) as well as survival analysis (MethSurv databases). (**A**,**B**) A box plot of two potential genes transcripts and beta values in normal (n = 114) and primary tumor (BRCA) tissue (n = 1097). Statistical significance was indicated by *t*-test *** and *p* < 0.0001 (left). Individual cancer stages are displayed in box plots (middle). The Kaplan–Meier curves represent the survival analysis of methylated genes in breast cancer (right).

**Figure 5 jpm-15-00134-f005:**
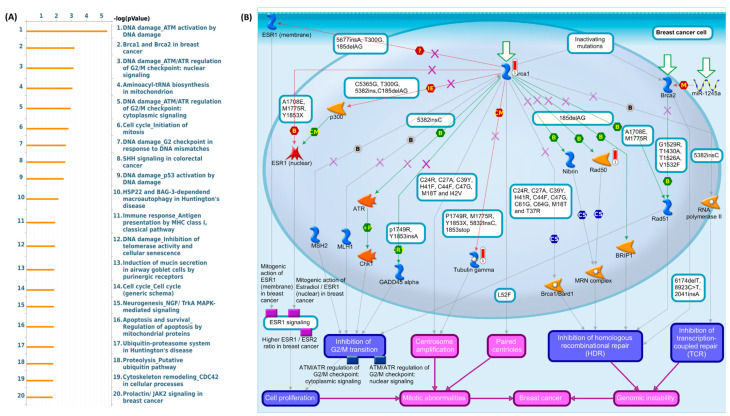
The expression of the GNPDA1 signaling pathway in breast cancer (MetaCore). This platform analyzed 10% of co-expressed genes that are correlated with GNPDA1 with a Spearman partial rho of ≥0.3. (**A**) Pathway distribution. We found that BRCA1 and BRCA2 were correlated with breast cancer development (with *p* < 0.05 as a cutoff value). (**B**) The signaling pathway of BRCA1 and BRCA2 in patients with breast cancer.

**Figure 6 jpm-15-00134-f006:**
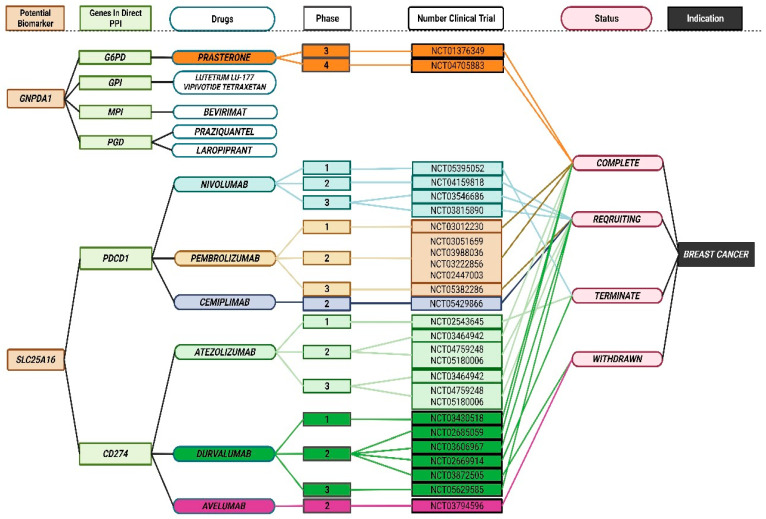
Connection between potential biomarkers of breast cancer genes, protein–protein interactions (PPIs) genes, and drugs available for breast cancer.

**Table 1 jpm-15-00134-t001:** Demographic Characteristics.

		Groups		
Variables	Total (n = 3060)	FHBC (n = 1676)	Non-FHBC (n = 1384)	*p* Value
	n (%)	n (%)	n (%)	
** *Biological factors* **				
Age (M ± SD)	51.2 ± 8.2	49.7 ± 7.2	52.7 ± 9.1	**<0.001**
BMI (M ± SD)	23.5 ± 3.7	23.5 ± 3.7	23.5 ± 3.7	0.994
Age at menarche (M ± SD)	13.3 ± 1.4	13.2 ±1.4	13.4 ± 1.5	**0.040**
Age at 1st live birth (M ± SD)	25.6 ± 6.1	24.5 ± 8.4	26.8 ±3.8	**<0.001**
First-degree relatives:				
Breast cancer mother				
No	2230 (72.9)	846 (27.6)	1384 (45.2)	**<0.001**
Yes	830 (27.1)	830 (27.1)	0 (0.0)	
Breast cancer sisters				
No	2185 (71.4)	801 (26.2)	1384 (45.2)	**<0.001**
Yes	875 (28.6)	875 (28.6)	0 (0.0)	
Has been pregnant				
No	383 (12.5)	212 (6.9)	171 (5.6)	0.821
Yes	2671 (87.5)	1462 (47.9)	1209 (39.6)	
Age at 1st pregnancy (M ± SD)	25.6 ± 8.5	25.5 ± 4.5	25.6 ± 3.9	0.487
Birth experience				
No	468 (15.3)	259 (8.5)	209 (6.8)	0.797
Yes	2585 (84.7)	1414 (46.3)	1171 (38.4)	
Breastfeeding practice				
Never	260 (8.5)	260 (8.5)	0 (0.0)	**<0.001**
No	1089 (35.6)	494 (16.1)	595 (19.5)	
Yes	1711 (55.9)	922 (30.1)	789 (25.8)	
** *Psychological and behavioral factors* **				
Depression status				
No	2899 (94.9)	1590 (52.1)	1309 (42.8)	0.945
Yes	156 (5.1)	86 (2.8)	70 (2.3)	
Alcohol consumption				
Never/occasionally	2988 (97.6)	1633 (53.4)	1355 (44.3)	**0.011**
Not recently	17 (0.6)	5 (0.2)	12 (0.4)	
Recently	55 (1.8)	38 (1.2)	17 (0.6)	
Smoking				
No	2760 (90.2)	1490 (48.7)	1270 (41.5)	**0.008**
Yes	300 (9.8)	186 (6.1)	114 (3.7)	
Secondhand smoke				
No	2842 (92.9)	1548 (50.6)	1294 (42.3)	0.197
Yes	217 (4.1)	128 (4.2)	89 (2.9)	
Exercise				
No	1783 (58.3)	1011 (33.1)	772 (25.2)	**0.011**
Yes	1276 (41.7)	664 (21.7)	612 (20.0)	
** *Social factors* **				
Marital Status				
Unmarried	344 (11.3)	197 (6.4)	147 (4.8)	0.334
Married	2713 (88.7)	1479 (48.4)	1234 (40.4)	
Educational level				
Primary school	337 (11.0)	180 (5.9)	157 (5.1)	0.590
High school or above	2722 (89.0)	1496 (48.9)	1226 (40.1)	
Job experience				
No	2160 (70.6)	1286 (42.0)	874 (28.6)	**<0.001**
Yes	900 (29.4)	390 (12.7)	510 (16.7)	
Live alone				
No	2777 (90.8)	1540 (50.3)	1237 (40.4)	**0.020**
Yes	282 (9.2)	136 (4.4)	146 (4.8)	

Notes: BMI (body mass index); FHBC (family history of breast cancer).

**Table 2 jpm-15-00134-t002:** List of Potential Genes.

No	Potential Genes	DEMs				DEmiRNAs		
		CpG sites	Location	*p* Value	FDR	hsa-miRNA	*p* Value	FDR
1	***GNPDA1* ***	**cg22647996**	TSS1500	0.000476001	0.639025617	** *hsa-miR-23a-3p* **	5.88 × 10^−37^	1.19 × 10^−34^
2	*ADARB2*	cg20205188	Body	0.000931185	0.891526514	hsa-miR-654-3p	2.51 × 10^−19^	1.70 × 10^−17^
3	*STOX2*	cg27457427	Body	0.0000935	0.255342042	hsa-miR-22-3p	9.18 × 10^−20^	6.66 × 10^−18^
4	*SHROOM2*	cg23553400	Body	0.000206531	0.4024522855	hsa-miR-23a-3p	5.88 × 10^−37^	1.19 × 10^−34^
5	*NCAPD3*	cg14934141	Body	0.000345934	0.542808906	hsa-miR-654-3p	2.51 × 10^−19^	1.70 × 10^−17^
6	*CCKBR*	cg26313599	Body	0.000435502	0.613241497	hsa-miR-654-3p	2.51 × 10^−19^	1.70 × 10^−17^
7	*ADAM19*	cg19464247	Body	0.005936274	1	hsa-miR-23a-3p	5.88 × 10^−37^	1.19 × 10^−34^
8	***SLC25A16* ***	**cg26546862**	TSS1500	0.004554406	1	** *hsa-miR-425-5p* **	3.79 × 10^−38^	9.63 × 10^−36^
9	*DLGAP2*	cg21498547	3′UTR	0.011187133	1	hsa-miR-22-3p	9.18 × 10^−20^	6.66 × 10^−18^
10	*ZNF787*	cg26951705	Body	0.017288423	1	hsa-miR-423-5p	8.53 × 10^−19^	4.81 × 10^−17^
11	*MRT04*	cg05704942	Body	0.046991025	1	hsa-miR-654-3p	2.51 × 10^−19^	1.70 × 10^−17^

* Genes were revealed from overlapping between DEM genes and DEmiRNA located in promotor sites as potential genes; FDR (false discovery rate).

## Data Availability

The data presented in this study are available on request from the corresponding author. The data are not public for privacy reasons.

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
