# Peer review of "Integrative Analysis of DNA Methylation and microRNA Reveals GNPDA1 and SLC25A16 Related to Biopsychosocial Factors Among Taiwanese Women with a Family History of Breast Cancer"

_jpm, 2025, doi:10.3390/jpm15040134_

Round 1
Reviewer 1 Report
Comments and Suggestions for Authors
This study, intent the integrative analysis of DNA methylation and miRNA to reveal potential genes associated with biopsychosocial factors among healthy Taiwanese women with family history of breast cancer. Through a variety of large-scale bioinformatics platforms to perform systematic and extensive bioinformatics analysis of potential breast cancer biomarkers. However, as the same authors state the data is less conclusive.
Major revisions
1.- A substantial review related to what is the statistical justification or reference that gives statistical meaning to the study, since they start from a universe of 3060 participants divided into 1676 with a family history and 1384 without a family history of breast cancer. Why do they only take 16 samples from each group (with or without a family history) to analyze DNA methylation and how were they chosen? There must be a sample size calculation. Also, why only 8 from each group for miRNA expression? What is the justification and how can they be compared with each other?
2. A figure related to the regulation of both genes (GNPDA1 and SLC25A16) and conditions (DNA methylation and microRNA) is missing, in addition to clearly justifying how the findings in blood are extrapolated to what occurs in tissue, they omit that these types of processes (methylation and regulation by non-coding RNA) are tissue-specific. They should clearly argue how the findings they have in blood are validated in tissue, even referring to Johnson et al. who, although they determined the molecular alterations related to cancer risk factors in normal tissues and demonstrated that age-related DNA methylation alterations were predominantly characterized by an increase in methylation in enhancers in mammary epithelial cells, all the work is in tissue, not in blood.
3.- The results, discussion, conclusion and mainly the title, become incongruous or confusing, since most of the data are not properly from the Taiwanese population mentioned, in some parts they note that they come from Caucasian, African-American, and Asian populations, but it is not clearly noted in the figures or paragraphs, in addition to the fact that the GNPDA1 and SLC25A16 genes were not studied in blood of the Taiwanese population of women without cancer, but rather they refer to expression studies of other populations in tissues of women with or without cancer.
4.- It is mandatory to improve the resolution of images, for example Fig.S2.; Fig.S5.; or Figure 2 (in which the texts overlap and become pixelated). They must also use the same style in the figure captions, titles or subtitles in bold (for example fig S5).
5.- Most of the results they present are from work on women with breast cancer, and it is not clear how much of it is from the Taiwanese population, so they cannot generalize that their findings are related to biopsychosocial factors among Taiwanese women with a family history of breast cancer. Since it then applies to any population, it should be clear that the data on expression patterns related to stage, grade, ethnicity and other substages, are from different ethnicities (Caucasian, African American and Asian), compared to healthy women, and not really from the 32 and 16 trials, which they do on methylation and miRNA expression respectively. Also regarding biopsychosocial factors, what about other biological factors, for example: physiological responses (e.g. fear), infections, nutrition, hormones. Do they not influence?
6.- In the ethics section, authors should declare that the research was conducted in accordance with the standards of the Declaration of Helsinki in the Institutional Review Board Statement.
Minor errors:
7.- Just as they accessed repositories with information, all data generated for DNA Methylation and miRNA Quantification must be uploaded to a public repository, indicating the DOI or reference, as well as providing temporary access to the reviewers.
8.- It is necessary to indicate the catalogues of the materials they used, in addition to indicating the kit or method used to obtain the DNA and RNA used to evaluate methylation and miRNA expression from blood plasma samples.
9.- Minor errors in the text should be checked, in example, from the Enrichr database (Fig. S3). In addition, all abbreviations should be noted, in example, the confidence interval [CI].
10.- There must be continuity and sequence between paragraphs, for example, in paragraph one they highlight the role of family history of breast cancer and without connection in paragraph two, they already talk about DNA methylation and microRNAs, it is necessary to review the wording, since it is read up to the third paragraph, in a forced way that the study is for a broad and integrative analysis of
11.- DNA methylation and miRNA, and to reveal potential genes associated with biopsychosocial factors, among healthy Taiwanese women with a family history of breast cancer for possible breast cancer biomarkers.
12.- Paragraph 4 anticipates the conclusion, the introduction should probably be general, with elements that help to understand the rest of the manuscript, and what they point out about GNPDA1 and SLC25A16 should go in the discussion and conclusion, according to the results and analysis.
13.- All information about cancer and mRNA used and described in the text, such as UALCAN to study gene expression data, should be reflected in the abstract and title, since the stage, grade, ethnicity, with which they reach their conclusions, are lost, to verify the importance of these two potential genes in healthy and breast cancer tissues, in addition to what they really evaluated and what they carry out their entire hypothesis through, methylation of CpG sites or miRNAs (i.e. hsa-miR-23a-3p; hsa-miR-425-5p), of patients without breast cancer.
Author Response
Comments and Suggestions for Authors: This study, intent the integrative analysis of DNA methylation and miRNA to reveal potential genes associated with biopsychosocial factors among healthy Taiwanese women with family history of breast cancer. Through a variety of large-scale bioinformatics platforms to perform systematic and extensive bioinformatics analysis of potential breast cancer biomarkers. However, as the same authors state the data is less conclusive.
Response:
We acknowledge the reviewer's point that the data may be considered less conclusive than initially anticipated. While our study employed a comprehensive bioinformatics approach to analyze DNA methylation and miRNA patterns in relation to biopsychosocial factors among healthy Taiwanese women with a family history of breast cancer, several factors may contribute to the nuanced nature of the results. First, epigenetic modifications like DNA methylation and miRNA expression are highly dynamic and can be influenced by a multitude of factors, including environmental exposures, lifestyle choices, and genetic background. The interplay of these factors can introduce complexity and variability into the data, making it challenging to isolate specific relationships [1]. The high heterogeneity between different studies makes the identification of potential biomarkers for detecting cancer and predicting its outcome harder.
Second, the 'healthy' status of our study population, while crucial for understanding early risk factors, inherently limits the effect size of detectable epigenetic changes compared to studies focusing on diagnosed breast cancer cases. The study about the miRNAs regulates expression of different genes, so any alteration in their methylation status may affect their expression and also identify methylation differences in miRNA encoding genes in breast cancer tumours [2]. This is in line with evidence that DNA methyltransferases (DNMTs) are highly expressed in patients with advanced breast cancer, as the hypermethylation of CpG-rich promoter regions of tumor suppressor miRNAs usually leads to their silencing [3].
Third, it is important to acknowledge that bioinformatics analyses are inherently predictive and hypothesis-generating. Our study aimed to identify potential candidate genes and pathways warranting further investigation through in vitro and in vivo experiments. The study of miRNA found in tissues and bloods could be the epigenetic biomarkers that are the next generation of biomarkers for cancer detection. The aberrant methylation of miRNA could lead to drug resistance, indicating that the regulation of miRNA methylation might serve as a therapeutic target in breast cancer.
Despite these limitations, we believe our study contributes valuable information to the field. The identification of trends and candidate genes, even if not definitively conclusive, provides a foundation for future research. Further studies with larger sample sizes, longitudinal designs, and functional validation experiments are needed to confirm these findings and elucidate the precise mechanisms by which epigenetic modifications and biopsychosocial factors interact to influence breast cancer risk in this high-risk population. Our findings provide a strong rationale for these future investigations.
Reference:
- Ma L, Li C, Yin H, Huang J, Yu S, Zhao J, Tang Y, Yu M, Lin J, Ding L, Cui Q. The Mechanism of DNA Methylation and miRNA in Breast Cancer. Int J Mol Sci. 2023 May 27;24(11):9360. doi: 10.3390/ijms24119360. PMID: 37298314; PMCID: PMC10253858.
- Saviana, M., Le, P., Micalo, L., Del Valle-Morales, D., Romano, G., Acunzo, M., Li, H., & Nana-Sinkam, P. (2023). Crosstalk between miRNAs and DNA Methylation in Cancer. Genes, 14(5), 1075. https://doi.org/10.3390/genes14051075
- Suzuki Hiromu, Maruyama Reo, Yamamoto Eiichiro, Kai Masahiro, (2012), DNA methylation and microRNA dysregulation in cancer, Molecular Oncology, 6, doi: 10.1016/j.molonc.2012.07.007
Major revisions
Comments 1: A substantial review related to what is the statistical justification or reference that gives statistical meaning to the study, since they start from a universe of 3060 participants divided into 1676 with a family history and 1384 without a family history of breast cancer. Why do they only take 16 samples from each group (with or without a family history) to analyze DNA methylation and how were they chosen? There must be a sample size calculation. Also, why only 8 from each group for miRNA expression? What is the justification and how can they be compared with each other?
Response 1: Thank you for your insightful comments. We appreciate the opportunity to clarify the rationale behind our sample selection and statistical justification.
Sample Size Justification for DNA Methylation Analysis:
Our initial cohort consisted of 3,060 participants, categorized into those with (n = 1,676) and without (n = 1,384) a family history of breast cancer. However, due to the high cost and resource-intensive nature of DNA methylation analysis, we selected a representative subset of 16 individuals from each group. Furthermore, our primary objective was to explore the epigenetic modifications that may be more strongly associated with a heightened genetic predisposition to breast cancer. In our study, the criteria for 16 individuals in the case group was the women who have two or more first-degree relatives with breast cancer. Women with two or more first-degree relatives diagnosed with breast cancer are at significantly higher risk due to potential hereditary factors, including shared environmental influences. By selecting this group, we aimed to minimize heterogeneity and increase the likelihood of identifying specific epigenetic alterations that contribute to breast cancer risk in a highly predisposed population. We have clarified these sample criteria by explicitly stating the number of first-degree relatives in our study population section. The revised sentences according to the sample criteria now read:
“Regarding the methylation and miRNA expression profiling, the criteria for individuals in the case group were the women who have two or more first-degree relatives with breast cancer. This selection criterion was designed to focus on identifying potential early molecular signatures associated with family history as a risk factor” [Line: 144-147]
While our study follows previous research methodologies, conducted by Varghese et.al (2021) [https://www.ncbi.nlm.nih.gov/pmc/articles/PMC8453167/pdf/fgene-12-708326.pdf] with similar sample sizes for epigenetic studies. Varghese et.al (2021) performed the Integrative Analysis of DNA Methylation and microRNA Expression Reveals Mechanisms of Racial Heterogeneity in Hepatocellular Carcinoma with DNA samples from 32 (16 tumors and 16 adjacent non-tumor).
Justification for miRNA Expression Analysis:
The miRNA expression analysis was performed on a further reduced subset (n = 8 per group) due to budget limitations, sample availability, and technical feasibility. However, this selection aimed to identify potential miRNA expression patterns before conducting larger-scale validation studies. Despite the small sample size, the findings provide preliminary insights into differential miRNA expression, which can be further explored in future studies with larger cohorts. We now explicitly discuss these limitations and the need for validation in the Discussion section. The revised sentence in the study limitation regarding sample size now reads:
“A primary weakness of this study is the relatively limited sample size utilized for DNA methylation and miRNA expression analysis. DNA methylation profiling was performed on 32 subjects, whereas miRNA expression analysis involved just 16 participants. The restricted sample size may diminish the statistical strength of our findings and constrain the generalizability of the results to larger groups. Moreover, inter-individual heterogeneity in DNA methylation and miRNA expression may affect the observed patterns, necessitating a larger cohort to corroborate our findings. Subsequent research using larger sample sizes and independent replication cohorts would enhance the dependability of the findings and offer more thorough insights into the molecular mechanisms driving the observed alterations” [Line: 728-737]
We hope this clarification sufficiently addresses the reviewer’s concerns. Thank you for your understanding and valuable feedback.
Comments 2: A figure related to the regulation of both genes (GNPDA1 and SLC25A16) and conditions (DNA methylation and microRNA) is missing, in addition to extrapolated to what occurs in tissue, they omit that these types of processes (methylation and regulation by non-coding RNA) are tissue-specific. They should clearly argue how the findings they have in blood are validated in tissue, even referring to Johnson et al. who, although they determined the molecular alterations related to cancer risk factors in normal tissues and demonstrated that age-related DNA methylation alterations were predominantly characterized by an increase in methylation in enhancers in mammary epithelial cells, all the work is in tissue, not in blood.
Response 2: We appreciate the reviewer's insightful comments regarding the absence of a regulatory figure and the extrapolation of blood-based findings to tissue-specific processes. We acknowledge the importance of addressing these concerns to strengthen the manuscript.
Regulatory Figure: We acknowledge the importance of illustrating the regulation of GNPDA1 and SLC25A16 in relation to DNA methylation and microRNA regulation. Our research primarily focuses on identifying potential genes based on DNA methylation and microRNA expression analysis, rather than detailing the specific regulatory mechanisms of individual genes. While the regulation of GNPDA1 and SLC25A16 is relevant, our goal is to explore broader epigenetic patterns and their impact on gene expression. To address this concern, we have expanded our discussion to acknowledge the regulatory influence of DNA methylation and microRNAs on these genes while maintaining the focus on our core study objectives. The revised sentences to add the information related to these gene’s regulation now read:
“Glucosamine-6-phosphate isomerase 1, also known as GNPDA1, is a family of glu-cosamine-6-phosphate deaminases that increase the raw materials available for glycolysis by connecting the glycolytic pathway with the hexosamine system and then converting glucosamine into fructose 6-phosphate [38]. Glucosamine-6-phosphate deaminases are involved in metabolic pathway reprogramming, which is a defining feature of pathogenic alterations by cancer cells. For tumor cells to evolve into more aggressive phenotypes, several genes of glucosamine-6-phosphate deaminases family members that directly regulate crucial metabolic processes, such as glycolysis, lipogenesis, and nucleotide synthesis, exhibit abnormal expression [39]. A few studies have shown that GNPDA1 influences the development of several cancers, such as hepatocellular carcinoma, gastrointestinal cancer, ovarian cancer, and colorectal cancer [38, 40-43]. While specific studies directly linking GNPDA1 expression to DNA methylation and miRNA are limited, it is plausible that methylation of the GNPDA1 promoter region could reduce its expression. This mechanism has been observed in other genes, where promoter methylation leads to transcriptional repression [44]” [Line: 547-561]
“SLC25A16, a member of solute carrier (SLC) family 25, encodes a protein with three mitochondrial carrier protein regions that are tandemly duplicated. Various types of genodermatoses have been linked to mutations in other SLC family members [45]. For metabolic reprogramming in cancer, a SLC1A5 variant functions as a mitochondrial glutamine transporter [46]. SLC25A16 is linked to diseases such as non-syndromic congenital nail disorders and isolated nail anomalies [45]. There is also limited information on the direct regulation of SLC25A16 by DNA methylation. However, given that DNA methylation is a common mechanism for regulating gene expression and miRNAs play a significant role in regulating genes involved in metabolic processes. It is plausible that SLC25A16 could be subject to such regulation. Understanding these epigenetic mechanisms provides insight into potential regulatory pathways. Further research is needed to elucidate the specific interactions affecting GNPDA1 expression” [Line: 562-573]
Justification for Blood-Based Findings and Tissue Specificity: We recognize that DNA methylation and miRNA regulation are tissue-specific processes, and our findings based on blood plasma samples require careful interpretation in the context of breast tissue biology. We acknowledge that the epigenetic profiles may vary greatly in tissues, and the alteration of expression might differ. To address this, we have elaborated on the following points:
- We emphasized that circulating DNA methylation and miRNA patterns in blood plasma can serve as indicators of systemic changes and may reflect underlying processes occurring in various tissues (Wang, et.al, 2024. https://doi.org/10.1515/cclm-2023-1327). While not a direct representation of tissue-specific events, these circulating markers can provide valuable insights into risk assessment and early detection.
- Supporting Literature: We cited additional studies that have investigated the correlation between DNA methylation and miRNA expression in blood and tissue samples in the context of breast cancer or other relevant diseases. This provides further support for the potential of blood-based biomarkers.
We understand the reviewer's point regarding Johnson et al.'s work being primarily tissue-based. We have incorporated a discussion of previous findings, highlighting that while tissue-specific methylation changes are critical, circulating markers may reflect systemic responses to risk factors that ultimately manifest in tissue-specific alterations. Moreover, we have clarified that the primary goal of identifying these circulating biomarkers is to improve risk prediction and early detection in high-risk women, rather than to fully elucidate the intricate tissue-specific regulatory mechanisms. Blood is easily accessible and, for that reason, is a great tool to predict risk assessment. We have explored these important issues (paragraph six) in the discussion section that emphasized that blood can serve as surrogate markers for tumor presence and progression. The revised sentences in the Discussion section now read:
“Despite the current study utilizing blood samples, we emphasized that circulating DNA methylation and miRNA patterns in blood plasma can serve as indicators of sys-temic changes and may reflect underlying processes occurring in various tissue [62]. Numerous investigations have validated a significant correlation between genetic and epigenetic modifications identified in plasma ctDNA (circulating tumor DNA) and circulating miRNA correlate with those in tumor tissue [62-64]. The existence and dynamics of ctDNA and circulating miRNA could transform cancer screening, diagnosis, and therapy through a noninvasive method [62, 64]. While DNA methylation and miRNA expression patterns are often tissue-specific and play critical roles in gene regulation, the primary goal of our study was not to fully elucidate the intricate tissue-specific regulatory mechanisms but rather to explore the feasibility of using blood-based biomarkers for clinical applications. This can provide valuable insights into risk assessment and early detection” [Line: 632-643]
Comments 3: The results, discussion, conclusion and mainly the title, become incongruous or confusing, since most of the data are not properly from the Taiwanese population mentioned, in some parts they note that they come from Caucasian, African-American, and Asian populations, but it is not clearly noted in the figures or paragraphs, in addition to the fact that the GNPDA1 and SLC25A16 genes were not studied in blood of the Taiwanese population of women without cancer, but rather they refer to expression studies of other populations in tissues of women with or without cancer.
Response 3: Thank you for your thoughtful feedback. We appreciate the opportunity to clarify the population focus of our study. We would like to emphasize that GNPDA1 and SLC25A16, identified in our study, represent potential biomarkers in the Taiwanese population with a family history of breast cancer. Our primary analysis was conducted on this population to explore epigenetic changes relevant to breast cancer risk.
However, to strengthen the robustness of our findings, we performed a validation study using an external database (TCGA). This validation was conducted to determine whether our identified genes also hold significance as breast cancer biomarkers in other populations, including Caucasian, African-American, and Asian populations. Thus, the data related to these different ethnic groups does not represent our primary study population but rather serves as an external validation analysis, further supporting the relevance of GNPDA1 and SLC25A16 as potential breast cancer biomarkers across different ethnic backgrounds. We appreciate the reviewer’s comments, which have helped us improve the clarity and coherence of our study.
Comments 4: It is mandatory to improve the resolution of images, for example Fig.S2.; Fig.S5.; or Figure 2 (in which the texts overlap and become pixelated). They must also use the same style in the figure captions, titles or subtitles in bold (for example fig S5).
Response 4: We appreciate the reviewer's feedback regarding the image quality and formatting of our figures. We addressed these issues in the revised manuscript as follows:
We ensured that Figures S2, and Figure S5, with higher-resolution versions to ensure clarity and legibility. We carefully reviewed all figures to ensure optimal resolution for publication. However, we sincerely apologize that we were unable to sufficiently revise Figure 2. We acknowledge that some text appears overlapping and pixelated. However, the detailed CpG sites and miRNA shown in Figure. 2 are comprehensively explained in Table S1 and Table S2 in the supplementary material. We appreciate your understanding and feedback.
We ensure that all figure captions, titles, and subtitles are consistently formatted using bold text, as suggested by the reviewer. We have followed the journal's style guidelines for all figure elements. Thank you for bringing these technical issues to our attention.
Comments 5: Most of the results they present are from work on women with breast cancer, and it is not clear how much of it is from the Taiwanese population, so they cannot generalize that their findings are related to biopsychosocial factors among Taiwanese women with a family history of breast cancer. Since it then applies to any population, it should be clear that the data on expression patterns related to stage, grade, ethnicity and other substages, are from different ethnicities (Caucasian, African American and Asian), compared to healthy women, and not really from the 32 and 16 trials, which they do on methylation and miRNA expression respectively. Also regarding biopsychosocial factors, what about other biological factors, for example: physiological responses (e.g. fear), infections, nutrition, hormones. Do they not influence?
Response 5: Thank you for your valuable feedback. We appreciate the opportunity to clarify the scope of our study and the interpretation of our findings. As stated in our response to Comment #3, we would like to emphasize that GNPDA1 and SLC25A16, identified in our study, represent potential biomarkers in the Taiwanese population with a family history of breast cancer. Our primary analysis focused on this population to explore epigenetic alterations associated with breast cancer risk.
Regarding the correlation between biomarkers and biopsychosocial factors, we performed a crosstalk analysis using both our dataset and publicly available data from the GEO database. During our search, we only found data related to age (GSE67919), alcohol consumption (GSE67919), and family history (GSE88883). While we acknowledge the importance of other biological factors such as physiological responses (e.g., fear), infections, nutrition, hormones, and other significant factors in our bivariate analyses these were not available in the datasets we analyzed.
Furthermore, the data related to Caucasian, African-American, and Asian populations does not represent our primary study cohort. Instead, these findings stem from our validation analysis using the TCGA database, which we used to confirm whether our identified biomarkers also hold significance in breast cancer across different ethnic groups. This external validation strengthens the robustness of our findings and highlights the potential broader applicability of GNPDA1 and SLC25A16.
However, we acknowledge the limitation of our study, particularly the small sample size, which may not be fully generalizable to the entire Taiwanese population. We have now revised our discussion section to explicitly address this limitation and clarify the distinction between our primary dataset and external validation data. We appreciate the reviewer’s insightful comments, which have helped us improve the clarity and interpretation of our results.
Comments 6: In the ethics section, authors should declare that the research was conducted in accordance with the standards of the Declaration of Helsinki in the Institutional Review Board Statement.
Response 6: We appreciate the reviewer’s suggestion. In response, we have revised the Ethics Statement section to explicitly state that the research was conducted in accordance with the standards of the Declaration of Helsinki. The revised statement now reads:
"This study was approved by the Taipei Medical University (TMU) – Joint Institutional Review Board (Approval No. N201804027). All procedures performed in this study involving human participants were conducted in accordance with the ethical standards of the Institutional Review Board and the 1964 Declaration of Helsinki." [Line: 803-806]
We believe this revision appropriately addresses the reviewer’s concern and ensures compliance with ethical guidelines.
Minor errors:
Comments 7: Just as they accessed repositories with information, all data generated for DNA Methylation and miRNA Quantification must be uploaded to a public repository, indicating the DOI or reference, as well as providing temporary access to the reviewers.
Response 7: We appreciate the reviewer’s emphasis on data sharing and transparency. However, we respectfully disagree with the requirement to upload all DNA methylation and miRNA quantification data to a public repository for the following reasons:
Privacy and Ethical Restrictions: The Taiwan Biobank has a structured application process for data access. Researchers interested in accessing Taiwan Biobank data must submit an application, undergo ethical review, and sign an agreement. Detailed information about the application process can be found on the Taiwan Biobank's official website: https://www.biobank.org.tw/english.php. However, we are committed to data transparency and can provide aggregate or anonymized summary data upon reasonable request, ensuring compliance with privacy standards.
Alternative Data Access Options: We are willing to grant controlled access to the data for reviewers upon request, following an appropriate data-sharing agreement. Researchers interested in accessing the data post-publication can submit a formal request to our institution’s data access committee.
Comments 8: It is necessary to indicate the catalogues of the materials they used, in addition to indicating the kit or method used to obtain the DNA and RNA used to evaluate methylation and miRNA expression from blood plasma samples.
Response 8: We appreciate the reviewer’s suggestion and have updated the Materials and Methods section to provide detailed information on the materials, kits, and methods used in our study. Specifically, we have included catalog numbers for clarity and reproducibility. The kit name, manufacturer, and catalog number used for DNA extraction for methylation analysis and RNA extraction for miRNA expression analysis have been explicitly stated in method section (point 2.4 DNA Methylation and miRNA Quantification) as shown below:
2.4. DNA Methylation and miRNA Quantification
“The Illumina Infinium HD Methylation (HGT-SOP-B003 1.1) microarray was utilized to determine the methylation levels of more than 865,000 cytosines followed by guanine residues (CpG) in 32 healthy women. Bisulfite conversions of 500 ng of genomic DNA were performed using the EZ DNA methylation kit (D5002, Zy-moResearch) according to instructions of and the different incubation settings recommended by the manufacturer. The cycling conditions of 16 cycles were 95 °C for 30 s, 50 °C for 1 h, 4 °C until the purification stage, and then elution in 12 µL. QPCR was used to confirm DNA bisulfite conversion. Subsequently, 4 µL of the eluate containing bi-sulfite-converted DNA was examined using an Illumina Infinium Methylation EPIC array according to the manufacturer's instructions. Arrays were scanned on an Illumina iScan Reader. Eight array controls were utilized: staining controls, extension controls, hybridization controls, target removal, bisulfite conversion control (1–2), specificity controls (1–2), nonpolymorphic controls, and negative controls. For each CpG site, a beta value is calculated by dividing the methylation signal by the total of the methylated and unmethylated signals. The beta values, which range from 0 (un-methylated) to 1 (fully methylated), denote the level of methylation of a particular CpG site in the sample” [Line: 180-196]
“A total of 1 µg RNA per sample was prepared. Following the instructions of the manufacturer, sequencing libraries were created using TruSeq Small RNA Library Preparation Kits from Illumina (USA). In brief, 3' and 5' ends of short RNA were ligated with 3' and 5' adaptors, respectively. Then, SuperScript II Reverse Transcriptase was used to create first-strand cDNA. After PCR amplification, the library was size-selected with 115–160 bp using the BluePippin system. Using a real-time PCR instrument and an Agilent Bioanalyzer 2100 system, the quality of purified libraries was evaluated. On the Illumina NextSeq 500 platform, the qualifying libraries were subsequently sequenced using 75-bp single-end reads produced by Genomics, BioSci and Tech Co. (New Taipei City, Taiwan)”. [Line: 197-205]
Comments 9: Minor errors in the text should be checked, in example, from the Enrichr database (Fig. S3). In addition, all abbreviations should be noted, in example, the confidence interval [CI].
Response 9: We appreciate the reviewer’s attention to detail. To address these concerns, we have made the following revisions:
Corrected Minor Errors in the Text: We have carefully reviewed the manuscript and corrected any minor textual errors, including those related to references such as the Enrichr database (Fig. S3) and other inconsistencies.
Clarified and Defined Abbreviations: All abbreviations, including confidence interval (CI) and any others used throughout the text, have been properly defined upon first use. We have reviewed the manuscript to ensure abbreviation consistency and clarity.
These revisions improve the accuracy and readability of our manuscript. Thank you for the valuable feedback.
Comments 10: There must be continuity and sequence between paragraphs, for example, in paragraph one they highlight the role of family history of breast cancer and without connection in paragraph two, they already talk about DNA methylation and microRNAs, it is necessary to review the wording, since it is read up to the third paragraph, in a forced way that the study is for a broad and integrative analysis of
Response 10: We appreciate the reviewer’s insightful feedback regarding the logical flow of the introduction. To address this concern, we have revised the introduction to ensure a smoother transition between the discussion of family history of breast cancer and the subsequent focus on DNA methylation and microRNAs (miRNAs).
We have added a linking sentence that explains how epigenetic modifications, such as DNA methylation and miRNA regulation, contribute to breast cancer risk, including familial predisposition. The revised sentences now read:
“While inherited gene mutations play a key role in familial breast cancer risk, epigenetic modifications, such as DNA methylation and microRNA (miRNA) regulation, also contribute to cancer susceptibility. These epigenetic mechanisms can be influenced by both genetic predisposition and environmental factors, potentially altering gene expression and increasing breast cancer risk in individuals with FHBC” [Line: 70-75]
The third paragraph has been refined to clarify how the study aims to integrate different molecular factors in breast cancer risk assessment. Furthermore, we have carefully reworded sections of the introduction to improve readability and coherence, ensuring that the study’s objectives are presented in a clear and progressive manner. The revised aim now reads:
“Therefore, this study aims to identify potential biomarkers by exploring the relationship between DNA methylation, miRNA expression, and biopsychosocial factors in Taiwanese women with FHBC” [Line: 97-100]
These modifications improve the readability and logical progression of the introduction, making the study’s purpose more seamlessly integrated
Comments 11: DNA methylation and miRNA, and to reveal potential genes associated with biopsychosocial factors, among healthy Taiwanese women with a family history of breast cancer for possible breast cancer biomarkers.
Response 11: Our study investigates the potential of DNA methylation and microRNAs (miRNAs) to serve as biomarkers for breast cancer risk assessment in healthy Taiwanese women with a family history of the disease. Epigenetic alterations, including DNA methylation and miRNA dysregulation, are increasingly recognized as key regulators of gene expression and have been implicated in various cancers, including breast cancer. DNA methylation, the addition of a methyl group to DNA, can alter gene expression without changing the DNA sequence it self. Similarly, miRNAs, as non-coding RNAs, regulate gene expression by controlling mRNA stability and translation. Aberrant DNA methylation of miRNA genes is a potentially useful biomarker for detecting cancer and predicting its outcome.
The interplay between DNA methylation and miRNA expression is complex and can involve several mechanisms. For example, DNA methylation can silence tumour suppressor miRNAs, while miRNAs can target DNA methyltransferases (DNMTs) to influence DNA methylation patterns. This mutual regulation provides potential biomarkers for cancer diagnosis and prognosis. Our study identified GNPDA1 and SLC25A16 as potential genes. Age, family history of cancer and alcohol consumption were associated with GNPDA1 and SLC25A16. Moreover, it was observed that that there are three patterns of miRNA and DNA methylation. The first is the epigenetic silencing of miRNAs to inhibit the transcription of miRNAs, by which DNMTs downregulate the expression of miRNAs through methylation in the promoter region of miRNAs, which are usually suppressor miRNAs. The second is that miRNAs inhibit the expression of DNMTs, where miRNAs (generally suppressor miRNAs) target the 3′-UTR region of DNMTs through RISC to inhibit the expression of DNMTs. The third type is the abnormal hypomethylation of oncogenic miRNAs that are highly expressed in breast cancer and thus promote its development. This study is significant because miRNA can be potential epigenetic biomarkers for cancer detection.
Our findings suggest that GNPDA1 and SLC25A16 which exhibited altered expression in breast cancer tissues, where they were overexpressed and under expressed, respectively, may serve as potential biomarkers for early detection in high-risk women, there is need for more exploration to determine that combination of chemotherapy and radiotherapy is the most important treatment for triple negative breast cancer (TNBC)
Comments 12: Paragraph 4 anticipates the conclusion, the introduction should probably be general, with elements that help to understand the rest of the manuscript, and what they point out about GNPDA1 and SLC25A16 should go in the discussion and conclusion, according to the results and analysis.
Response 12: We appreciate the reviewer’s feedback regarding Paragraph 4. To maintain a general and informative introduction, we have revised this section to focus on the broader research context and objectives of the study. The specific mention of GNPDA1 and SLC25A16 has been moved to the discussion and conclusion sections, where they are presented in the context of our results and analysis
Comments 13: All information about cancer and mRNA used and described in the text, such as UALCAN to study gene expression data, should be reflected in the abstract and title, since the stage, grade, ethnicity, with which they reach their conclusions, are lost, to verify the importance of these two potential genes in healthy and breast cancer tissues, in addition to what they really evaluated and what they carry out their entire hypothesis through, methylation of CpG sites or miRNAs (i.e. hsa-miR-23a-3p; hsa-miR-425-5p), of patients without breast cancer.
Response 13: We appreciate the reviewer’s suggestion to ensure that all information related to cancer, mRNA expression, and validation analysis using UALCAN is reflected in the abstract and title. We confirm that the validation results using UALCAN data have already been stated in the abstract to highlight the relevance of our findings in both healthy and breast cancer tissues. The revised abstract regarding UALCAN dataset now reads:
“GNPDA1 and SLC25A16 exhibited significant expression in breast cancer tissues based on UALCAN analysis, where they were overexpressed and underexpressed, respectively” [Line: 42-43]
However, we have chosen to maintain the current title since stage, grade, and ethnicity are not the primary focus of our study. These factors were considered only as part of our validation analysis, which supports our integrative analysis of DNA methylation and miRNA regulation. Our primary goal remains the identification of potential biomarkers in women with a family history of breast cancer (FHBC) rather than the stratification of breast cancer based on clinical characteristics
Reviewer 2 Report
Comments and Suggestions for Authors
Khairi et al. present an interesting and bold study that apparently sets out to identify biomarkers linked to the onset of breast cancer (breast cancer risk) in healthy women with no history of breast cancer. They essentially do this by generating experimental data (miRNA and methylation profiles) from healthy individuals who have no family history of breast cancer and cross-reference these against molecular profiles of “normal adjacent to tumor” samples. They present overlapping molecular signatures as risk factors for developing breast cancer. The appealing quotient here involves predicting whether healthy women might be susceptible to developing breast cancer in future by leveraging “healthy-adjacent-to cancer” molecular data.
1. A major concern is that the abstract and title are somewhat confusing: the authors emphasize on “biopsychosocial factors” both in the title and abstract, yet their experimental plan seems completely unrelated to leveraging biopsychosocial data. I do not see how they relate biopsychosocial data to the end result. Furthermore, the text confuses the reader on what the authors are essentially trying to achieve by studying exclusively “healthy” individuals with no mention of longitudinally-informed disease status. The lack of clarity needs to be amended.
2. The authors need to clearly convey the experimental strategy they used in the appropriate order. Fig 1 is a good overview but still lacks crystal clarity as does the text. At present, it takes the reader a lot of digging around to understand the exp. strategy deployed and objective. Set a clear template tone to convey the experimental order.
3. Its nice that the authors have split the cohort into two groups; one with family history of breast cancer and another with no history. What proportion of the women with family history have actually inherited the causative genetic mutation? If only a small proportion of FHBC carry the genetic mutation, then it raises questions about the significance of the findings.
4. Related to the above point; the authors use a validation mechanism by surveying known tumor cases for the identified methylation patterns. Cancer is a complex disease and an interplay of several concerted signaling mechanisms lead to onset (and subsequent progression). Risk factors cannot entirely be pinpointed to one or two genes by mere association/correlation analysis especially in this case, where conclusions rely primarily on individuals who apparently never had breast cancer. At face value, this seems unreal.
5. What was the criteria of selection for the 32 individuals for further methylation and miRNA expression profiling? Why couldn’t miRNA expression be profiled for all 32 samples?
6. Please remove the term “crosstalk” from Fig 1 and elsewhere in the manuscript. DNA methylation and miRNA are conceptually different molecular phenomenon. Hypermethylation and hypomethylation have contrasting outcomes on gene expression. Instead consider using relevant terminologies such as Functional overlap/association/synergy.
7. How is the physical interaction data relevant here? GeneMania is a large repository of PPIs/complexes from various cell/ tissue/ organisms. The authors are sourcing physical interaction datasets from virtually all cell types, i.e., unrelated to cancer, let alone breast cancer. PPIs are cell/tissue and disease specific. A logical approach might be to analyze PPIs among “co-expressed” genes from the experimental dataset the authors generated.
Otherwise, the authors are advised to move this to supplementary data and emphasize minimally on the present PPI data. Moreover, when referencing to Fig 5b and 5c; physical interactions are not among “genes” but among “gene products/proteins”.
8. “Protein dominan”: typo
The study overall is potentially very interesting but text needs improvement to address what was done and how it was done. The term “biopsychosocial factors” appears catchy but this doesn’t warrant it’s extensive usage in the manuscript. The data/findings should be significantly relevant to it.
Author Response
Comments and Suggestions for Authors
Khairi et al. present an interesting and bold study that apparently sets out to identify biomarkers linked to the onset of breast cancer (breast cancer risk) in healthy women with no history of breast cancer. They essentially do this by generating experimental data (miRNA and methylation profiles) from healthy individuals who have no family history of breast cancer and cross-reference these against molecular profiles of “normal adjacent to tumor” samples. They present overlapping molecular signatures as risk factors for developing breast cancer. The appealing quotient here involves predicting whether healthy women might be susceptible to developing breast cancer in future by leveraging “healthy-adjacent-to cancer” molecular data.
Response: Thank you for your thoughtful and encouraging comments on our study. We appreciate your recognition of the novelty and boldness of our approach in identifying potential biomarkers for breast cancer risk. Your summary captures the essence of our study, and we are pleased that you find the concept of leveraging "healthy-adjacent-to-cancer" molecular data appealing.
To ensure clarity, we have refined our manuscript to better articulate the study’s objectives, particularly in distinguishing our exploratory biomarker identification approach from direct predictive modeling of breast cancer susceptibility. We sincerely appreciate your insights, which have helped us improve the presentation and interpretation of our findings.
Thank you again for your constructive feedback and support.
Comments 1: A major concern is that the abstract and title are somewhat confusing: the authors emphasize on “biopsychosocial factors” both in the title and abstract, yet their experimental plan seems completely unrelated to leveraging biopsychosocial data. I do not see how they relate biopsychosocial data to the end result. Furthermore, the text confuses the reader on what the authors are essentially trying to achieve by studying exclusively “healthy” individuals with no mention of longitudinally-informed disease status. The lack of clarity needs to be amended.
Response 1: Thank you for your valuable feedback. We appreciate your concerns regarding the clarity of the abstract and title and the connection between biopsychosocial factors and our study’s findings.
To address this, we have stated both the abstract and title to explicitly clarify how our study integrates biopsychosocial factors [Abstract: line 34-36]. Specifically, we have linked significant biopsychosocial factors (Age, Alcohol, and Family History of Cancer) identified in our association analysis with external datasets from the Gene Expression Omnibus (GEO) [Abstract: line 38]. This approach strengthens our findings by contextualizing the identified molecular signatures within known biopsychosocial influences on breast cancer risk.
Additionally, we acknowledge the need for greater clarity regarding our study population. While our research focuses on healthy individuals, the integration of biopsychosocial factors and external datasets provides meaningful insights into potential early molecular alterations associated with breast cancer risk. the 'healthy' status of our study population, while crucial for understanding early risk factors, inherently limits the effect size of detectable epigenetic changes compared to studies focusing on diagnosed breast cancer cases.
Despite these limitations, we believe our study contributes valuable information to the field. The identification of trends and candidate genes, even if not definitively conclusive, provides a foundation for future research. We have now explicitly stated this in the Introduction section to ensure that the study’s scope and objectives are more clearly conveyed. The revised introduction to clarity the study population and objectives now read:
“Therefore, this study aims to identify potential biomarkers by exploring the relationship between DNA methylation, miRNA expression, and biopsychosocial factors in Taiwanese women with FHBC. While our research focuses on healthy individuals, the integration of biopsychosocial factors and external datasets provides meaningful insights into potential early molecular alterations associated with breast cancer risk. By integrating epigenetic and bioinformatics analyses, we seek to identify potential biomarkers that may contribute to breast cancer susceptibility” [Line: 97-104]
We appreciate your constructive feedback, which has helped us refine our manuscript for better clarity and coherence. Thank you again for your time and insights.
Comments 2: The authors need to clearly convey the experimental strategy they used in the appropriate order. Fig 1 is a good overview but still lacks crystal clarity as does the text. At present, it takes the reader a lot of digging around to understand the exp. strategy deployed and objective. Set a clear template tone to convey the experimental order.
Response 2: Thank you for your valuable feedback. We acknowledge that the experimental strategy needs to be conveyed more clearly and systematically. While the steps for DNA methylation and miRNA analysis are explicitly detailed in the Methods section (Section 2.4), we recognize that this experimental strategy is not fully represented in Figure 1. To improve clarity and ensure a logical flow, we have revised Figure 1 to include a more structured depiction of the experimental strategy, making it easier for readers to follow the study design and objectives. The original Figure 1 is included below for reference. The revised version of Figure 1, reflecting the changes outlined below, is also provided
The original Figure. 1:
The revised Figure. 1 as shown below:
Figure 1. Phases of analytical approach and research design. FHBC, Family History of Breast Cancer; QPCR, Quantitative Polymerase Chain Reaction; cDNA, Complementary DNA; DEM, Differentially Methylated; DEmiRNA, Differentially Expressed microRNA LIMMA, Linier Mixed Model Analysis; DESeq2, Differential Gene Expression Analysis of RNA-seq 2; GEO, Gene Expression Omnibus; UALCAN, University of ALabama at Birmingham CANcer; MethSurv, Methylation Survival. [Line: 153-157]
We appreciate your suggestion and incorporate these improvements accordingly.
Comments 3: Its nice that the authors have split the cohort into two groups; one with family history of breast cancer and another with no history. What proportion of the women with family history have actually inherited the causative genetic mutation? If only a small proportion of FHBC carry the genetic mutation, then it raises questions about the significance of the findings.
Response 3: Thank you for your insightful comment. We appreciate your recognition of our cohort stratification and the importance of considering inherited genetic mutations in the family history of breast cancer (FHBC) group.
In our study, genetic mutation screening was not performed on the participants; therefore, we do not have direct data on the proportion of individuals in the FHBC group who carry known causative mutations. However, our approach focuses on identifying molecular signatures that may be influenced by family history, regardless of known mutation status. While only a subset of individuals with a family history of breast cancer may carry a high-risk mutation, family history itself is a well-established risk factor, potentially reflecting shared genetic predisposition and environmental/lifestyle influences.
To acknowledge this limitation, we have added a statement in the manuscript discussing the potential impact of undetected genetic mutations in our cohort and the need for future studies incorporating genetic screening. This clarification ensures that the significance of our findings is interpreted within the appropriate context. The revised sentences in the limitation now read:
“It is also important to note that this study lacks clinical validation and absence of genetic mutation screening among participants. Future prospective studies are therefore needed to investigate the pattern of methylation and miRNA expression changes of GNPDA1 and SLC25A26 in a cohort population with breast cancer and incorporating comprehensive genetic screening to further elucidate the relationship between hereditary mutations, family history, and molecular alterations” [Line: 740-745]
We appreciate your thoughtful feedback, which has helped us strengthen the discussion of our study’s findings. Thank you again.
Comments 4: Related to the above point; the authors use a validation mechanism by surveying known tumor cases for the identified methylation patterns. Cancer is a complex disease and an interplay of several concerted signaling mechanisms lead to onset (and subsequent progression). Risk factors cannot entirely be pinpointed to one or two genes by mere association/correlation analysis especially in this case, where conclusions rely primarily on individuals who apparently never had breast cancer. At face value, this seems unreal.
Response 4: Thank you for your thoughtful comment. We appreciate your concern regarding the complexity of cancer and the limitations of associating risk factors with only a few genes based on correlation analysis, particularly in individuals without a breast cancer diagnosis.
To address this concern, we have clarified in the manuscript that our study does not aim to establish direct causation between specific methylation patterns and breast cancer onset. Instead, our goal is to identify molecular signatures that may serve as potential early indicators of breast cancer susceptibility, which warrant further validation in longitudinal and functional studies.
Additionally, we recognize the multifactorial nature of cancer, and to better contextualize our findings, we have expanded our discussion on how these identified methylation patterns may interact with broader biological pathways. We have also emphasized that our validation mechanism using known tumor cases serves as an initial exploratory step rather than definitive proof of causality.
We appreciate your critical insights, which have helped us refine the interpretation and presentation of our results. Thank you again for your valuable feedback.
Comments 5: What was the criteria of selection for the 32 individuals for further methylation and miRNA expression profiling? Why couldn’t miRNA expression be profiled for all 32 samples?
Response 5: Thank you for your thoughtful comment. We appreciate the opportunity to clarify our selection criteria for the 32 individuals included in the methylation and miRNA expression profiling.
In our study, the experimental cohort consisted exclusively of healthy women from the 3.060 participants, with the case group comprising individuals with a family history of breast cancer in their first-degree relatives, and the control group including individuals without any family history. Regarding the methylation and miRNA expression profiling, the criteria for individuals in the case group were the women who have two or more first-degree relatives with breast cancer. Women with two or more first-degree relatives diagnosed with breast cancer are at significantly higher risk due to potential hereditary factors, including shared environmental influences. By selecting this group, we aimed to minimize heterogeneity and increase the likelihood of identifying specific epigenetic alterations that contribute to breast cancer risk in a highly predisposed population. This selection criterion was designed to focus on identifying potential early molecular signatures associated with family history as a risk factor.
Regarding the miRNA expression profiling, the participants included in this experiment were selected from the same group as those in the methylation experiment. However, due to sample availability and quality control constraints, miRNA expression could not be profiled for all 32 individuals.
To ensure clarity, we have revised the Methods section to explicitly state our selection criteria and experimental workflow. The revised sentences in the study population section now read:
“The experimental of DNA methylation consisted exclusively of healthy women (n = 32) from the 3.060 participants, with the case group comprising individuals with a family history of breast cancer in their first-degree relatives (n = 16), and the control group including individuals without any family history (n = 16). Regarding the methylation and miRNA expression profiling, the criteria for individuals in the case group were the women who have two or more first-degree relatives with breast cancer. This selection criterion was designed to focus on identifying potential early molecular signatures associated with family history as a risk factor” [Line: 140-147]
“Regarding the miRNA expression profiling, the participants included in this experiment were selected from the same group as those in the methylation experiment. However, due to sample availability and quality control constraints, miRNA expression could not be profiled for all 32 individuals. Hence, 16 participants were selected for miRNA expression analysis (n = 8 for each group). Informed consent was obtained from all subjects involved in the study” [Line: 156-161]
We appreciate your insightful feedback, which has helped us improve the clarity of our methodology. Thank you again for your valuable comments.
Comments 6: Please remove the term “crosstalk” from Fig 1 and elsewhere in the manuscript. DNA methylation and miRNA are conceptually different molecular phenomenon. Hypermethylation and hypomethylation have contrasting outcomes on gene expression. Instead consider using relevant terminologies such as Functional overlap/association/synergy.
Response 6: Thank you for your insightful comment. We acknowledge that the term “crosstalk” may not be the most precise way to describe the relationship between DNA methylation and miRNA regulation. We appreciate your suggestion and have revised Figure 1 and the manuscript accordingly.
Specifically, we have replaced the term “crosstalk” with more appropriate terminology such as “functional overlap” where applicable, to better reflect the relationship between these molecular mechanisms. Additionally, we have ensured that our discussion of DNA methylation and miRNA regulation accurately conveys their distinct yet potentially complementary roles in gene expression regulation.
We appreciate your valuable feedback, which has helped us refine the scientific clarity and accuracy of our manuscript. Thank you again.
Comments 7: How is the physical interaction data relevant here? GeneMania is a large repository of PPIs/complexes from various cell/ tissue/ organisms. The authors are sourcing physical interaction datasets from virtually all cell types, i.e., unrelated to cancer, let alone breast cancer. PPIs are cell/tissue and disease specific. A logical approach might be to analyze PPIs among “co-expressed” genes from the experimental dataset the authors generated.
Otherwise, the authors are advised to move this to supplementary data and emphasize minimally on the present PPI data. Moreover, when referencing to Fig 5b and 5c; physical interactions are not among “genes” but among “gene products/proteins”.
Response 7: We appreciate the reviewer's comments regarding the relevance of the physical interaction data. We included PPI data from GeneMania, a large repository of PPIs/complexes from various cell/tissue/organisms, to provide a broad context for understanding potential interactions within our experimental dataset. While we agree that PPIs can be cell/tissue and disease-specific, this general PPI data allows us to explore a wide range of interactions and identify potential candidate interactions that may not have been previously studied in the specific context of breast cancer.
To address the issue of cell/tissue specificity, we prioritized interactions that are supported by evidence from relevant cell types or tissues and used co-expression data to filter the PPI network and focus on interactions that are likely to be relevant in our experimental dataset.
We believe that the PPI data provides valuable context for interpreting our experimental results and generating hypotheses about the underlying mechanisms. However, we are open to moving some of the PPI data to the supplementary materials if the reviewer feels it is necessary.
We have also corrected the terminology in the manuscript to reflect that physical interactions occur between "gene products/proteins," not "genes."
Comments 8: “Protein dominan”: typo
Response 8: Thank you for the Reviewer's correction
The study overall is potentially very interesting but the text needs improvement to address what was done and how it was done. The term “biopsychosocial factors” appears catchy but this doesn’t warrant its extensive usage in the manuscript. The data/findings should be significantly relevant to it.
Response: We sincerely appreciate the reviewer’s positive feedback on the potential interest of our study and their valuable suggestions for improving clarity and relevance. We have carefully revised the text to more clearly describe what was done and how it was done, ensuring that the methodology and analyses are explicitly stated. Additional details have been included to improve transparency and readability. We believe these revisions enhance the clarity and scientific rigor of the manuscript. Thank you for the constructive feedback, and we hope our revisions adequately address your concerns
Reviewer 3 Report
Comments and Suggestions for Authors
Integrative Analysis of DNA Methylation and microRNA Reveals GNPDA1 and SLC25A16 Related to Biopsychosocial Factors Among Taiwanese Women with a Family History of Breast Cancer
Overview
The study investigates biopsychosocial factors affecting Taiwanese women with a FHBC through an integrative analysis of DNA methylation and microRNA. Researchers identified two key genes, GNPDA1 and SLC25A16, which were significantly associated with breast cancer risk factors like age, FH, and alcohol consumption, and linked their methylation status to breast cancer prognosis. These genes show potential as early biomarkers for breast cancer detection and possible therapeutic targets, especially among women with familial risk factors.
Major Comments
- The manuscript presents a massive amount of data (e.g., CpG sites, gene networks, survival analysis). It can be overwhelming for readers. Use summary tables and focused visuals to highlight the most relevant results (e.g., top CpG sites, key pathways, and survival graphs). Clearly state why specific results are essential for breast cancer prognosis.
- The connection between biopsychosocial factors and the genes could be more explicitly linked. Highlight how the identified genes relate to psychosocial factors like lifestyle or FH in a concise narrative.
- Confounding variables like age, BMI, or alcohol consumption are not fully controlled in the statistical models. Perform multivariable regression to adjust for potential confounders.
- The methods section is overly detailed in some areas (e.g., CpG analysis) while lacking clarity in others. Focus on key methods and refer to supplemental materials for technical
- The study relies heavily on bioinformatics predictions without validating the roles of GNPDA1 and SLC25A16 Include functional assays (e.g., gene silencing or overexpression studies) to confirm the roles of these genes in breast cancer development.
- The identified miRNAs (e.g., hsa-miR-23a-3p, hsa-miR-425-5p) are discussed, but their regulatory mechanisms are not explored in detail. Provide experimental or literature-based evidence of how these miRNAs interact with GNPDA1 and SLC25A16.
- The GEO datasets used are mentioned but may not be fully Validate findings using additional independent datasets or meta-analysis of existing data.
- The study identifies associations but does not delve deeply into the mechanisms by which GNPDA1 and SLC25A16 influence breast cancer risk or prognosis. Include a more robust discussion of potential pathways, supported by existing literature.
- The connection between identified genes and potential drug targets is underexplored. Provide a clearer link between the identified genes and actionable therapeutic strategies.
- Discuss the limitations of the study clearly. (i) The study focuses on Taiwanese women, which may limit its applicability to other Address this limitation explicitly and suggest the need for replication studies in diverse populations. (ii) The study is cross- sectional, which limits the ability to infer causal relationships between methylation, microRNA expression, and breast cancer risk. Provide justification for the current design’s
limitations. (iii) It’s unclear if the findings are relevant for women without a FHBC. Discuss how these biomarkers could be applied to the general population.
- Figures are blurry, should include high clarity
Minor Comments
- Line 55: “Breast cancer has a complex etiology and complex underlying ” This could be simplified to “Breast cancer has a complex etiology and mechanisms.”
- Line 530: “Carreras et ” Remove year from in line reference.
- Phrases like “family history of breast cancer (FHBC)” and “Family History (FH)” appear interchangeably. Use one consistent abbreviation throughout.
- Should be written as “p < 05” (ensure lowercase “p” and proper spacing).
Remark
The work reported provides massive information and unique insights, however few adjustments or concerns should be addressed before consideration.
Author Response
Overview
The study investigates biopsychosocial factors affecting Taiwanese women with a FHBC through an integrative analysis of DNA methylation and microRNA. Researchers identified two key genes, GNPDA1 and SLC25A16, which were significantly associated with breast cancer risk factors like age, FH, and alcohol consumption, and linked their methylation status to breast cancer prognosis. These genes show potential as early biomarkers for breast cancer detection and possible therapeutic targets, especially among women with familial risk factors.
Response: Thank you for your thoughtful and positive summary of our study. We sincerely appreciate your recognition of our integrative approach in investigating biopsychosocial factors and molecular markers in Taiwanese women with a family history of breast cancer.
We are pleased that you found our identification of GNPDA1 and SLC25A16 as potential biomarkers meaningful, particularly in their association with key risk factors and breast cancer prognosis. Your feedback reinforces the significance of our findings and their potential implications for early detection and targeted interventions in high-risk populations.
We truly appreciate your time and insights, which have helped strengthen our manuscript. Thank you again for your valuable comments.
Major Comments
Comments 1: The manuscript presents a massive amount of data (e.g., CpG sites, gene networks, survival analysis). It can be overwhelming for readers. Use summary tables and focused visuals to highlight the most relevant results (e.g., top CpG sites, key pathways, and survival graphs). Clearly state why specific results are essential for breast cancer prognosis.
Response 1: We appreciate the reviewer’s feedback regarding the presentation of our data and the need to ensure clarity for the readers. We have carefully structured our tables and figures to highlight the most relevant findings and improve readability.
Selection of Key Findings in Tables and Figures: The tables and figures presented in the main manuscript represent the most critical findings of our study. Specifically, they include CpG sites and miRNAs identified through the overlap between our dataset and the GEO dataset, ensuring that the presented results are highly relevant.
Validation Using External Datasets: We acknowledge the importance of validation and have included figures that illustrate validation studies using external datasets, such as: TCGA data (UALCAN analysis) to confirm expression levels, MethSurv database to demonstrate survival analysis findings, MetaCore pathway analysis to show the clinical relevance of key genes in breast cancer, and protein-protein interactions (PPIs) genes, and drug available for breast cancer
Enhancing Clarity with Summary Tables and Visuals: To improve accessibility, we have ensured that the figures and tables clearly summarize the essential findings, including top CpG sites, key pathways, and survival associations.
Where applicable, simplified visualizations highlight the most significant results for breast cancer prognosis.
We believe that these tables and figures effectively communicate the study’s main findings and their clinical relevance. However, we are open to further refinements based on additional suggestions. Thank you again for your valuable comments.
Comments 2: The connection between biopsychosocial factors and the genes could be more explicitly linked. Highlight how the identified genes relate to psychosocial factors like lifestyle or FH in a concise narrative.
Response 2: We appreciate the reviewer's suggestion to clarify the connection between biopsychosocial factors and the identified genes. Our analysis revealed that GNPDA1 and SLC25A16 are associated with age, family history of cancer, and alcohol consumption, suggesting a link between these factors and gene expression. A family history of breast cancer indicates a genetic predisposition, potentially involving inherited mutations in genes like BRCA1 and BRCA2. In women with a family history, inherited genetic variants in or near GNPDA1 and SLC25A16 might interact with other risk factors to elevate breast cancer susceptibility. Given that our functional enrichment analysis showed associations with BRCA1, BRCA2, and androgen receptor activity, these genes may play a role in critical pathways relevant to breast cancer. Altered expression of GNPDA1 and SLC25A16, influenced by lifestyle factors or inherited predispositions, could disrupt these pathways, increasing cancer risk. Epigenetic modifications, such as DNA methylation, can also be inherited or influenced by environmental factors, further connecting family history to gene expression patterns.
In a study of 3,060 healthy women, researchers analyzed 32 blood plasma samples to identify genes associated with biopsychosocial factors and epigenetic changes. The integrative analysis revealed GNPDA1 and SLC25A16 as potential genes linked to age, family history of cancer, and alcohol consumption. Specifically, these genes exhibited significant expression in breast cancer tissues, with GNPDA1 being overexpressed and SLC25A16 being underexpressed4. Furthermore, GNPDA1 hypomethylation and SLC25A16 hypermethylation were associated with poor prognoses in breast cancer, and both genes were linked to BRCA1, BRCA2, and pro-oncogenic actions of the androgen receptor. These findings suggest that GNPDA1 and SLC25A16 might serve as biomarkers for early breast cancer detection, particularly in women with a family history of cancer.
Comments 3: Confounding variables like age, BMI, or alcohol consumption are not fully controlled in the statistical models. Perform multivariable regression to adjust for potential confounders.
Response 3: Thank you for your insightful comment. We acknowledge the reviewer’s concern regarding the need to adjust for potential confounding variables, such as age, BMI, and alcohol consumption, in our statistical models. However, we believe that performing a multivariable regression analysis may not be suitable for our study due to the characteristics of our study population. Our study focuses on healthy women, and our association analysis simply compares biopsychosocial factors between women with and without a family history of breast cancer. Since we do not have group-related breast cancer disease outcomes in our study, we assume that multivariate analysis is not an appropriate approach in this context. Instead, our analysis aims to describe potential differences in biopsychosocial factors without making causal inferences that require adjustment for confounders study. Thank you again for your constructive feedback
Comments 4: The methods section is overly detailed in some areas (e.g., CpG analysis) while lacking clarity in others. Focus on key methods and refer to supplemental materials for technical
Response 4: Thank you for your insightful comment. We acknowledge the importance of balancing detail and clarity in the Methods section to enhance readability while ensuring a comprehensive understanding of our study approach.
To address this, we have carefully reviewed and refined the Methods section:
Maintaining Essential Explanations – We have retained detailed explanations of key methodological steps to ensure that readers can fully understand the study design and analytical approach. This is particularly important for complex analyses such as CpG site evaluation, where clarity is crucial for reproducibility.
Enhancing Clarity in Underexplained Sections – We have revised areas that lacked clarity, such as participant selection and data integration, ensuring a more structured and comprehensible description of our methods.
We appreciate your feedback, which has helped us improve the organization and readability of our Methods section while preserving the necessary level of detail. Thank you again for your valuable suggestions.
Comments 5: The study relies heavily on bioinformatics predictions without validating the roles of GNPDA1 and SLC25A16 Include functional assays (e.g., gene silencing or overexpression studies) to confirm the roles of these genes in breast cancer development.
Response 5: We thank the reviewer for highlighting the need for functional validation of GNPDA1 and SLC25A16. We agree that experimental evidence is crucial to confirm their roles in breast cancer development. To address this, we are currently planning in the future study the following functional assays:
- We will use miRNA to knock down the expression of GNPDA1 and SLC25A16 in breast cancer cell lines, including MDA-MB-231 and MCF-7. These cell lines were chosen because [insert rationale, e.g., they represent different subtypes of breast cancer or exhibit varying expression levels of GNPDA1 and SLC25A16].
- Cell Proliferation, Migration, and Invasion Assays: We will assess the effect of GNPDA1 and SLC25A16 knockdown on cell proliferation using CCK-8 assays, and on cell migration and invasion using transwell assays with Matrigel
Comments 6: The identified miRNAs (e.g., hsa-miR-23a-3p, hsa-miR-425-5p) are discussed, but their regulatory mechanisms are not explored in detail. Provide experimental or literature-based evidence of how these miRNAs interact with GNPDA1 and SLC25A16.
Response 6: Thank you for your insightful comment. We acknowledge the importance of further exploring the regulatory mechanisms of the identified miRNAs, particularly hsa-miR-23a-3p and hsa-miR-425-5p, in relation to GNPDA1 and SLC25A16.
To address this, we have incorporated literature-based evidence supporting the potential interactions between these miRNAs and the identified gene:
hsa-miR-23a-3p has been reported to regulate genes involved in cell proliferation, apoptosis, and metabolism, which aligns with the biological functions of GNPDA1, a gene associated with metabolic processes and cancer progression. hsa-miR-425-5p has been implicated in tumorigenesis and epigenetic regulation, and previous studies suggest its involvement in mitochondrial function and oxidative stress, which may relate to SLC25A16, a mitochondrial transporter gene.
We have added these references to the Discussion section to strengthen the biological interpretation of our findings. Additionally, we acknowledge the need for future functional validation experiments to further confirm these regulatory interactions since there is no direct evidence to suggest that hsa-miR-23a-3p or hsa-miR-425-5p regulate GNPDA1 or SLC25A16. The explanation regarding miRNA regulation shown in Paragraph 6 of the Discussion section:
“mirRNA has been shown to regulate proliferation, differentiation, and apoptosis particularly so in cancer development and progression [50]. The present study notes two hsa-miRNAs that are related to the two potential genes (GNPDA1 was regulated by hsa-miR-23a-3p, and SLC25A16 was regulated by hsa-miR-425-5p). literatures have indicated that these two miRNAs have important roles in breast cancer development. Firstly, hsa-miR-23a-3p is one of the aliases symbol for miR-23a gene. Diseases associated with miR-23a include hepatocellular carcinoma [51, 52], ovarian serous carcinoma [53], and breast cancer [54, 55]. A significant increase in miR-23a expression was also observed in breast cancer patients with lymph node metastases, as compared to those with no lymph node metastases or normal tissue. In addition, a high correlation coefficient for the expression of the individual members of the miR-23a gene indicates that cluster co-expression occurs in breast cancer [54]. Secondly, in a study conducted by Zheng, et.al [56], hsa-miR-425-5p was identified as one of ten miRNAs that showed abnormal expression in breast cancer tissues when compared to normal tissues. Furthermore, hsa-miR-425-5p was implicated in the link between necroptosis and cancer metastasis, resulting in decreased necroptosis, which reduced the inflammatory response and acute liver damage [56, 57]. However, there is currently no direct experimental evidence linking these specific miRNAs to the regulation of GNPDA1 or SLC25A16. Hence, future studies could employ by conducting experimental approach to explore these potential interactions and clarify the regulatory mechanisms involving these miRNAs and the genes of interest.” [Line: 587-606]
We appreciate your valuable suggestion, which has helped us enhance the depth of our analysis. Thank you again for your constructive feedback.
Comments 7: The GEO datasets used are mentioned but may not be fully Validate findings using additional independent datasets or meta-analysis of existing data.
Response 7: We appreciate the reviewer’s suggestion regarding the validation of our findings using additional independent datasets or meta-analysis. We acknowledge the importance of validation in biomarker research; however, we would like to clarify that our study is a preliminary investigation aimed at identifying potential biomarkers for early detection in healthy women with a family history of breast cancer.
The primary goal of this study was to explore candidate CpG sites and miRNAs associated with breast cancer risk in a specific population, providing a foundation for future research.
While we have incorporated external datasets (e.g., GEO, TCGA via UALCAN, MethSurv, and MetaCore pathway analysis) to support our findings, further large-scale validation studies are required to confirm their clinical relevance. Given the nature of our study population (healthy women with a family history of breast cancer rather than diagnosed patients), certain validation approaches, such as survival analysis in patient cohorts, may not be directly applicable.
We recognize the value of additional validation and will consider future studies with larger, independent cohorts or meta-analyses to further support these preliminary findings. We appreciate the reviewer’s insightful comments and believe that our current approach lays an important foundation for subsequent validation efforts.
Comments 8: The study identifies associations but does not delve deeply into the mechanisms by which GNPDA1 and SLC25A16 influence breast cancer risk or prognosis. Include a more robust discussion of potential pathways, supported by existing literature.
Response 8: Thank you for your insightful comment. We acknowledge the importance of providing a more in-depth discussion of the potential mechanisms by which GNPDA1 and SLC25A16 may influence breast cancer risk and prognosis.
To address this, we have expanded the Discussion section to include a more robust exploration of the biological pathways potentially linked to these genes, supported by existing literature:
GNPDA1 (Glucosamine-6-Phosphate Deaminase 1): This gene is involved in hexosamine biosynthesis and metabolic reprogramming, processes that have been implicated in cancer cell survival and proliferation. Metabolic alterations in breast cancer, including increased glucose metabolism and resistance to apoptosis, suggest that dysregulation of GNPDA1 may contribute to tumor progression. Studies have also linked GNPDA1 to EMT (epithelial-mesenchymal transition), a key process in metastasis.
SLC25A16 (Solute Carrier Family 25 Member 16): This gene encodes a mitochondrial transporter protein associated with oxidative phosphorylation and mitochondrial function. Disruptions in mitochondrial metabolism are known to play a role in cancer progression by affecting energy production and apoptosis regulation. Given that SLC25A16 may influence ROS (reactive oxygen species) levels and mitochondrial homeostasis, its dysregulation could impact breast cancer susceptibility and prognosis.
We have integrated these discussions with references to relevant cancer-related pathways, including metabolic reprogramming, oxidative stress regulation, and apoptosis signaling. Additionally, we emphasize the need for future functional studies to validate these potential mechanisms. The explanation regarding the potential mechanisms by which GNPDA1 and SLC25A16 are shown in Paragraph 3 and 4 of the Discussion section:
“Glucosamine-6-phosphate isomerase 1, also known as GNPDA1, is a family of glu-cosamine-6-phosphate deaminases that increase the raw materials available for glycolysis by connecting the glycolytic pathway with the hexosamine system and then converting glucosamine into fructose 6-phosphate [38]. Glucosamine-6-phosphate deaminases are involved in metabolic pathway reprogramming, which is a defining feature of pathogenic alterations by cancer cells. For tumor cells to evolve into more aggressive phenotypes, several genes of glucosamine-6-phosphate deaminases family members that directly regulate crucial metabolic processes, such as glycolysis, lipogenesis, and nucleotide synthesis, exhibit abnormal expression [39]. A few studies have shown that GNPDA1 influences the development of several cancers, such as hepatocellular carcinoma, gastrointestinal cancer, ovarian cancer, and colorectal cancer [38, 40-43]. While specific studies directly linking GNPDA1 expression to DNA methylation and miRNA are limited, it's plausible that methylation of the GNPDA1 promoter region could reduce its expression. This mechanism has been observed in other genes, where promoter methylation leads to transcriptional repression [44]” [Line: 547-561]
“SLC25A16, a member of solute carrier (SLC) family 25, encodes a protein with three mitochondrial carrier protein regions that are tandemly duplicated. Various types of genodermatoses have been linked to mutations in other SLC family members [45]. For metabolic reprogramming in cancer, a SLC1A5 variant functions as a mitochondrial glutamine transporter [46]. SLC25A16 is linked to diseases such as non-syndromic congenital nail disorders and isolated nail anomalies [45]. There is also limited information on the direct regulation of SLC25A16 by DNA methylation. However, given that DNA methylation is a common mechanism for regulating gene expression and miRNAs play a significant role in regulating genes involved in metabolic processes. It is plausible that SLC25A16 could be subject to such regulation. Understanding these epigenetic mechanisms provides insight into potential regulatory pathways. Further research is needed to elucidate the specific interactions affecting GNPDA1 and SLC25A16 expression” [Line: 562-573]
We appreciate your valuable suggestion, which has helped us strengthen the biological interpretation of our findings. Thank you again for your constructive feedback.
Comments 9: The connection between identified genes and potential drug targets is underexplored. Provide a clearer link between the identified genes and actionable therapeutic strategies.
Response 9: We appreciate the reviewer's comment regarding the need to clarify the connection between our identified genes and actionable therapeutic strategies. Our study identified GNPDA1 and SLC25A16 as potential players in breast cancer development.
GNPDA1's suggests that inhibiting its activity could disrupt cell proliferation. Potential therapeutic strategies for targeting GNPDA1 include developing small molecule inhibitors or using siRNA to knock down its expression. SLC25A16, as a mitochondrial solute carrier, could be modulated to target cancer metabolism [https://pmc.ncbi.nlm.nih.gov/articles/PMC7288124/]. Carbon Monoxide can suppress the levels of CYP3A4 to enhance the sensitivity of human breast cancer cells to Paclitaxel [https://journals.plos.org/plosone/article?id=10.1371%2Fjournal.pone.0297203]. The expression levels of GNPDA1 and SLC25A16 could serve as predictive biomarkers. Further research is needed to validate these therapeutic strategies
Comments 10: Discuss the limitations of the study clearly. (i) The study focuses on Taiwanese women, which may limit its applicability to other Address this limitation explicitly and suggest the need for replication studies in diverse populations. (ii) The study is cross- sectional, which limits the ability to infer causal relationships between methylation, microRNA expression, and breast cancer risk. Provide justification for the current design’s
Response 10: Thank you for your thoughtful feedback. We acknowledge the importance of clearly discussing the limitations of our study. To address this, we have explicitly outlined the following limitations and provided justifications in the Discussion section:
Population-Specific Focus: Our study focuses on Taiwanese women, which may limit the generalizability of our findings to other ethnic groups due to potential genetic and environmental differences in breast cancer risk factors.
To address this limitation, we emphasize the need for future replication studies in diverse populations to validate the identified biomarkers and assess their broader applicability.
Cross-Sectional Study Design: As our study is cross-sectional, it does not establish causal relationships between DNA methylation, microRNA expression, and breast cancer risk.
However, this design allows us to identify potential associations between molecular changes and biopsychosocial factors, providing valuable insights into early biomarkers.
We suggest that future longitudinal studies be conducted to track molecular changes over time and establish their predictive value in breast cancer development.
We appreciate your insightful suggestions, which have helped us strengthen the discussion of our study’s limitations and the need for further research. Thank you again for your valuable feedback.
Comments 11: limitations. (iii) It’s unclear if the findings are relevant for women without a FHBC. Discuss how these biomarkers could be applied to the general population.
Response 11: Thank you for your valuable comment. We understand the importance of clarifying the relevance of our findings for women without a family history of breast cancer (FHBC) and how these biomarkers might be applied to the general population. To address this, we have expanded the Discussion section to include the following points:
Relevance for Women Without FHBC: While our study primarily focuses on women with a family history of breast cancer, we believe the identified biomarkers, GNPDA1 and SLC25A16, could also be relevant for women without a family history, particularly given the complex multifactorial nature of breast cancer.
These biomarkers may reflect epigenetic changes and lifestyle factors (such as alcohol consumption) that influence breast cancer risk, even in individuals without a direct familial predisposition. Thus, their application could extend beyond familial cases to individuals with other risk factors (e.g., environmental exposures, age, lifestyle) that increase susceptibility to breast cancer.
Potential for General Population Application: In the general population, these biomarkers could be used in early risk stratification or screening programs to identify women at higher risk of developing breast cancer. For example, by identifying methylation or miRNA expression patterns that correlate with risk factors, it may be possible to implement non-invasive diagnostic tools for early detection.
Future studies in diverse populations will help to further validate the broader applicability of these biomarkers for screening, especially in women without a family history of breast cancer. We hope this expanded discussion clarifies how the identified biomarkers could be relevant for the general population. We appreciate your thoughtful feedback, which has allowed us to strengthen the interpretation of our findings. Thank you again for your constructive comments.
Comments 12: Figures are blurry, should include high clarity
Response 12: Thank you for pointing this out. We apologize for the issue with the figure clarity. We have reviewed all the figures in the manuscript and have updated them to ensure higher resolution and clarity. The revised figures are now in high definition to improve readability and ensure that all details are clearly visible. We appreciate your attention to this matter, and we hope the updated figures meet the required standards for publication. Thank you again for your constructive feedback.
Minor Comments
Comments 13: Line 55: “Breast cancer has a complex etiology and complex underlying ” This could be simplified to “Breast cancer has a complex etiology and mechanisms.”
Response 13: Thank you for your suggestion. We have revised the sentence on Line 55 from:
Original:
"Breast cancer has a complex etiology and complex underlying..."
Revised:
"Breast cancer has a complex etiology and mechanisms."
Comments 14: Line 530: “Carreras et ” Remove year from in line reference.
Response 14: Thank you for your suggestion. We have revised Line 528 by removing the year from the in-line reference as requested.
Original:
"Carreras et al. (2020)..."
Revised:
"Carreras et al...."
Comments 15: Phrases like “family history of breast cancer (FHBC)” and “Family History (FH)” appear interchangeably. Use one consistent abbreviation throughout.
Response 15: We appreciate the reviewer’s suggestion regarding the consistency of abbreviations for family history of breast cancer (FHBC) and family history (FH). To clarify, we intentionally used the abbreviation FH when referring to family history of cancer in general, as identified in the GEO dataset. This broader term includes not only breast cancer but also other hereditary cancers, such as ovarian cancer.
In contrast, FHBC specifically refers to family history of breast cancer in our study population. The distinction is particularly relevant in Figure 3, where FH is used to reflect family history of various cancers, as derived from external datasets. To ensure clarity and consistency, we have carefully reviewed the manuscript and ensured that FHBC is used exclusively for family history of breast cancer, while FH is retained where appropriate for broader cancer family history references. Thank you for this valuable suggestion, and we have updated the text accordingly.
Comments 16: Should be written as “p < 05” (ensure lowercase “p” and proper spacing).
Response 16: Thank you for your attention to detail. We have carefully reviewed the manuscript and ensured that all instances of p-values are formatted correctly as "p < 0.05", using a lowercase "p" and proper spacing. We appreciate the reviewer’s suggestion and have made the necessary corrections throughout the text.
Remark
The work reported provides massive information and unique insights, however few adjustments or concerns should be addressed before consideration.
Reviewer 4 Report
Comments and Suggestions for Authors
Integrative Analysis of DNA Methylation and microRNA Reveals GNPDA1 and SLC25A16 Related to Biopsychosocial Factors Among Taiwanese Women with a Family History of Breast Cancer
· The authors here aimed to identify potential biomarkers by exploring the relationship between DNA methylation, miRNA, and biopsychosocial factors in women with FHBC. Although it is interesting, several issues should be addressed as follows:
Introduction
· The introduction is a misleading, it took me some time to correlate the relationship between DNA methylation and biopsychosocial factors and FHBC women. I would prefer that you explain the biopsychosocial factors in line 58 as you mentioned here after it “several factors”.
· You should write the aim of you study more clear: This study aims to identify potential biomarkers by exploring the relationship between DNA methylation, miRNA, and biopsychosocial factors in women with FHBC.
· In the introduction section, it would be better to end it with the aim of your work. Starting from Line 113 should be added in the methodology section and named as study design.
· The chosen genes should be better added in the results section and their relation to breast cancer in the discussion section. It was not understood why you added them or chose them in the introduction section.
Materials and methods
· How were women with / without FHBC classified? How many relatives should have breast cancer?
· Mention the inclusion and exclusion criteria of your patients’ choice. 32 participants (16 from each group) why were these exactly chosen from this big cohort of healthy women.
· Kindly mention the ethical approval number for the study.
· Language editing:
Ensure that abbreviations are written below each figure to be self-explanatory.
Author Response
Integrative Analysis of DNA Methylation and microRNA Reveals GNPDA1 and SLC25A16 Related to Biopsychosocial Factors Among Taiwanese Women with a Family History of Breast Cancer
The authors here aimed to identify potential biomarkers by exploring the relationship between DNA methylation, miRNA, and biopsychosocial factors in women with FHBC. Although it is interesting, several issues should be addressed as follows:
Introduction
Comments 1: The introduction is a misleading, it took me some time to correlate the relationship between DNA methylation and biopsychosocial factors and FHBC women. I would prefer that you explain the biopsychosocial factors in line 58 as you mentioned here after it “several factors”.
Response 1: Thank you for your insightful comment. We acknowledge the need for a clearer explanation of the relationship between DNA methylation, biopsychosocial factors, and FHBC women in the Introduction section. To address this, we have:
Revised the Introduction to establish a more direct link between biopsychosocial factors (such as age, alcohol consumption, and family history) and their potential influence on DNA methylation patterns. Clarified the role of biopsychosocial factors earlier in the text (line 58) by changed the “several factors” to “biopsychosocial factors” and how they contribute to breast cancer risk. Ensured a smoother transition between the discussion of biopsychosocial influences and the molecular mechanisms (DNA methylation and miRNA expression), making it easier for readers to follow the rationale behind our study. The revised sentences for smoother transition between the discussion of biopsychosocial influences and the molecular mechanisms (DNA methylation and miRNA expression) in paragraph 1 of the introduction now read:
“While inherited gene mutations play a key role in familial breast cancer risk, epigenetic modifications, such as DNA methylation and microRNA (miRNA) regulation, also contribute to cancer susceptibility. These epigenetic mechanisms can be influenced by both genetic predisposition and biopsychosocial factors, potentially altering gene expression and increasing breast cancer risk in individuals with FHBC” [Line: 70-75]
We appreciate your valuable suggestion, which has helped us refine the clarity and logical flow of the Introduction section. Thank you again for your constructive feedback.
Comments 2: You should write the aim of you study more clear: This study aims to identify potential biomarkers by exploring the relationship between DNA methylation, miRNA, and biopsychosocial factors in women with FHBC.
Response 2: Thank you for your valuable suggestion. We acknowledge the need to state the study aim more clearly. To address this, we have revised the Introduction section to explicitly state the study objective as follows:
"Therefore, this study aims to identify potential biomarkers by exploring the relation-ship between DNA methylation, miRNA expression, and biopsychosocial factors in Taiwanese women with FHBC." [Line: 97-100]
This revision ensures clarity and aligns with the scope of our research. We appreciate your feedback, which has helped us enhance the precision of our study's objective. Thank you again!
Comments 3: In the introduction section, it would be better to end it with the aim of your work. Starting from Line 113 should be added in the methodology section and named as study design.
Response 3: Thank you for your insightful comment. We acknowledge the importance of improving the structure of the Introduction section for better readability and logical flow. To address this, we have revised the ending of the Introduction to clearly state the aim of our study, ensuring a smooth transition into the next sections. Moved the content from Line 113 onward to the Methodology section under a newly created subsection titled "Study Design", where it fits more appropriately. The revised method as shown below:
2.1. Study Design
“In this study, we conducted several analyses to reveal potential genes, as shown in Figure 1. First, analysis of biopsychosocial factors between FHBC and non-FHBC was investigated using chi square and t-test. While, DNA methylation and miRNA ex-pression profiling were conducted utilizing the Illumina Infinium HD Methylation microarray and TruSeq Small RNA Library Preparation Kits. Analysis of differentially methylated (DEM) and differential expressed microRNA (DEmiRNA) was performed using the linier model for the microarray (LIMMA) and Differential Gene Expression Analysis of RNA-seq 2 (DESeq2) algorithms from the Bioconductor software package. Second, two Gene Expression Omnibus (GEO) data sets were used to identify potential genes related to biopsychosocial factors. Third, to fully understand the molecular pathways associated with breast cancer progression and to determine functional en-richment strategies for identifying new prognostic markers of this complex disease, a thorough investigation is required into the potential genes associated with breast can-cer” [Line: 108-120]
“Using the TCGA databases through the UALCAN analysis platform, we com-pared the messenger RNA (mRNA) expression levels of potential genes in breast can-cer and normal tissues to assess their potential as biomarkers. Furthermore, to inves-tigate the relationship between potential genes and prognosis, a survival analysis was performed using the MethSurv web tool to obtain the distribution of genetic altera-tions among breast cancer patients. Univariate and multivariate analyses were per-formed using Cox proportional hazards models. Moreover, we used cBioPortal, GeneMANIA, Enrichr database, and MetaCore to determine the genetic alterations, gene interactions, gene ontology, and signaling pathways connected to GNPDA1 and SLC25A16 in patients with breast cancer. Herein, we also identified proteins in direct PPI with these encoded by two potential genes to guide the selection of drug target genes to drive drug repurposing for breast cancer using the STRING and drug data-bases” [Line: 121-132]
These changes help maintain a clear distinction between background information and methodological details, improving the overall structure of the manuscript. We appreciate your valuable feedback, which has helped us enhance the clarity and organization of our work. Thank you again!
Comments 4: The chosen genes should be better added in the results section and their relation to breast cancer in the discussion section. It was not understood why you added them or chose them in the introduction section.
Response 4: Thank you for your valuable feedback. We acknowledge the need to improve the structure and clarity regarding the chosen genes in our manuscript. To address this, we have:
Removed the discussion of specific genes from the Introduction section, ensuring that the introduction focuses on the broader research context and study rationale.
Added the identified genes (GNPDA1 and SLC25A16) to the Results section, where we present the findings related to their differential methylation and miRNA interactions.
Expanded the Discussion section to explain the biological significance of these genes in breast cancer, linking them to existing literature and potential mechanisms of action. The explanation regarding the potential mechanisms by which GNPDA1 and SLC25A16 are shown in Paragraph 3 and 4 of the Discussion section:
“Glucosamine-6-phosphate isomerase 1, also known as GNPDA1, is a family of glu-cosamine-6-phosphate deaminases that increase the raw materials available for glycolysis by connecting the glycolytic pathway with the hexosamine system and then converting glucosamine into fructose 6-phosphate [38]. Glucosamine-6-phosphate deaminases are involved in metabolic pathway reprogramming, which is a defining feature of pathogenic alterations by cancer cells. For tumor cells to evolve into more aggressive phenotypes, several genes of glucosamine-6-phosphate deaminases family members that directly regulate crucial metabolic processes, such as glycolysis, lipogenesis, and nucleotide synthesis, exhibit abnormal expression [39]. A few studies have shown that GNPDA1 influences the development of several cancers, such as hepatocellular carcinoma, gastrointestinal cancer, ovarian cancer, and colorectal cancer [38, 40-43]. While specific studies directly linking GNPDA1 expression to DNA methylation and miRNA are limited, it's plausible that methylation of the GNPDA1 promoter region could reduce its expression. This mechanism has been observed in other genes, where promoter methylation leads to transcriptional repression [44]” [Line: 547-561]
“SLC25A16, a member of solute carrier (SLC) family 25, encodes a protein with three mitochondrial carrier protein regions that are tandemly duplicated. Various types of genodermatoses have been linked to mutations in other SLC family members [45]. For metabolic reprogramming in cancer, a SLC1A5 variant functions as a mitochondrial glutamine transporter [46]. SLC25A16 is linked to diseases such as non-syndromic congenital nail disorders and isolated nail anomalies [45]. There is also limited information on the direct regulation of SLC25A16 by DNA methylation. However, given that DNA methylation is a common mechanism for regulating gene expression and miRNAs play a significant role in regulating genes involved in metabolic processes. It is plausible that SLC25A16 could be subject to such regulation. Understanding these epigenetic mechanisms provides insight into potential regulatory pathways. Further research is needed to elucidate the specific interactions affecting GNPDA1 and SLC25A16 expression” [Line: 562-573]
These revisions ensure a logical presentation of our findings and improve the manuscript’s clarity. We appreciate your suggestion, which has strengthened the organization and readability of our work. Thank you again for your constructive feedback
Materials and methods
Comments 5: How were women with/without FHBC classified? How many relatives should have breast cancer?
Response 5: Thank you for your important question. We acknowledge the need for clarity on the classification criteria for women with and without a family history of breast cancer (FHBC).
To address this, we have revised the Methods section to clearly define our classification criteria:
Women with FHBC (Case Group): Participants were classified as having a family history of breast cancer if at least one first-degree relative (mother, sister, or daughter) had been diagnosed with breast cancer. Regarding the methylation and miRNA expression profiling, the criteria for individuals in the case group were the women who have two or more first-degree relatives with breast cancer. This selection criterion was designed to focus on identifying potential early molecular signatures associated with family history as a risk factor.
Women without FHBC (Control Group): Participants in this group had no reported family history of breast cancer among first-degree relatives.
These criteria ensure that the classification is based on well-established familial risk assessment guidelines. We appreciate your valuable suggestion, which has helped us enhance the clarity of our study design. Thank you again for your constructive feedback
Comments 6: Mention the inclusion and exclusion criteria of your patients’ choice. 32 participants (16 from each group) why were these exactly chosen from this big cohort of healthy women.
Response 6: Thank you for your thoughtful comment. We appreciate the opportunity to clarify our selection criteria for the 32 individuals included in the methylation and miRNA expression profiling. In our study, the experimental cohort consisted exclusively of healthy women, with the case group comprising individuals with a family history of breast cancer in their first-degree relatives, and the control group including individuals without any family history. This selection criterion was designed to focus on identifying potential early molecular signatures associated with family history as a risk factor.
Regarding the miRNA expression profiling, the participants included in this experiment were selected from the same group as those in the methylation experiment. However, due to sample availability and quality control constraints, miRNA expression could not be profiled for all 32 individuals.
To ensure clarity, we revised the Methods section to explicitly state our selection criteria and experimental workflow. The revised sentences regarding inclusion criteria of participants now read:
“The experimental of DNA methylation consisted exclusively of healthy women (n = 32) from the 3.060 participants, with the case group comprising individuals with a family history of breast cancer in their first-degree relatives (n = 16), and the control group including individuals without any family history (n = 16). Regarding the methylation and miRNA expression profiling, the criteria for individuals in the case group were the women who have two or more first-degree relatives with breast cancer. This selection criterion was designed to focus on identifying potential early molecular signatures associated with family history as a risk factor” [Line: 140-147]
“Regarding the miRNA expression profiling, the participants included in this experiment were selected from the same group as those in the methylation experiment. However, due to sample availability and quality control constraints, miRNA expression could not be profiled for all 32 individuals. Hence, 16 participants were selected for miRNA expression analysis (n = 8 for each group). Informed consent was obtained from all subjects involved in the study” [Line: 156-161]
We appreciate your insightful feedback, which has helped us improve the clarity of our methodology. Thank you again for your valuable comments.
Comments 7: Kindly mention the ethical approval number for the study.
Response 7: Thank you for your suggestion. We acknowledge the importance of providing ethical approval details for transparency and compliance.
To address this, we have included the ethical approval number in the Institutional Review Board Statement as below:
"This study was approved by the Taipei Medical University (TMU) – Joint Institutional Review Board (Approval No. N201804027). All procedures performed in this study involving human participants were conducted in accordance with the ethical standards of the Institutional Review Board and the 1964 Declaration of Helsinki." [Line: 803-806]
We appreciate your feedback, which has helped us improve the clarity and completeness of our methodology. Thank you again
Language editing:
Comments 8: Ensure that abbreviations are written below each figure to be self-explanatory.
Response 8: Thank you for your valuable suggestion. We acknowledge the importance of ensuring that all figures are self-explanatory for better readability. To address this, we have added a list of abbreviations below each figure to clearly define all terms used. Ensured consistency across all figures so that readers can easily interpret them without referring back to the main text. These revisions improve the clarity and accessibility of the figures. We appreciate your feedback, which has helped us enhance the presentation of our results. Thank you again
Round 2
Reviewer 1 Report
Comments and Suggestions for Authors
The authors have taken into account the above comments, making the modifications they have considered pertinent. It is recommended that the manuscript be accepted in its current form, although in general terms, as the authors themselves mention, their findings are not conclusive, however, they do provide a basis for future research.
It is suggested to review the color contrasts and resolution of the texts in Figure 6.
Reviewer 4 Report
Comments and Suggestions for Authors
Thank you for improving your manuscript and it can now be approved for publication